# DynaPhArM: Adaptive and Physics-Constrained Modeling for Target-Drug Complexes with Drug-Specific Adaptations

**Diya Zhang[1], Mengwei Sun[1], Xingdan Wang[1], Cheng Liang[2]***
**Qiaozhen Meng[3]*, Shiqiang Ma[4]*, Fei Guo[1]***

[1]Central South University   [2]Shandong Normal University
[3]Xiangtan University   [4]Shenzhen Institutes of Advanced Technology, Chinese Academy of Sciences

{diya.zhang, 234711070, 244711083, guofei}@csu.edu.cn
ALCS417@sdnu.edu.cn
qiaozhenm@xtu.edu.cn
sq.ma@siat.ac.cn

## Abstract

Accurately modeling the target-drug complex at atom level presents a significant challenge in the computer-aided drug design. Traditional methods that rely solely on rigid transformations often fail to capture the adaptive interactions between targets and drugs, particularly during substantial conformational changes in targets upon ligand binding, which becomes especially critical when learning target-drug interactions in drug design. Accurately modeling these changes is crucial for understanding target-drug interactions and improving drug efficacy. To address these challenges, we introduce DynaPhArM, an SE(3)-Equivariant Transformer model specifically designed to capture adaptive alterations occurring within target-drug interactions. DynaPhArM utilizes the cooperative scalar-vector representation, drug-specific embeddings, and a diffusion process to effectively model the evolving dynamics of interactions between targets and drugs. Furthermore, we integrate physical information and energetic principles that maintain essential geometric constraints, such as bond lengths, bond angles, van der Waals forces (vdW), within a multi-task learning (MTL) framework to enhance accuracy. Experimental results demonstrate that DynaPhArM achieves state-of-the-art performance with an overall root mean square deviation (RMSD) of 2.01 Å and a sc-RMSD of 0.29 Å while exhibiting higher success rates compared to existing methodologies. Additionally, DynaPhArM shows promise in enhancing drug specificity, thereby simulating how targets adapt to various drugs through precise modeling of atomic-level interactions and conformational flexibility.

## 1 Introduction

Accurate modeling of the 3D structures of target-drug complexes provides essential insights into the molecular binding modes and interaction mechanisms between drugs and their target proteins [1, 2]. Furthermore, it enhances our understanding of inter-individual variability in drug metabolism and pharmacodynamic responses. The insights gained from simulating target-drug interactions establish a robust theoretical foundation for the development of personalized therapeutic strategies, thereby improving drug selectivity and specificity while minimizing adverse effects [3].

---

*Corresponding authors.

39th Conference on Neural Information Processing Systems (NeurIPS 2025).

Traditional methods, such as AutoDock [4], AutoDock Vina [5], ZDOCK [6] and RDOCK [7], often rely on rigid assumptions or static representations of target structures. These approaches fail to account for adaptive conformational changes. Such conformational alterations are not merely incidental. They are integral to the binding process, significantly influencing the strength, specificity and overall efficacy of interactions. Consequently, these methods become inadequate when targets undergo substantial conformational changes upon binding with various drugs. To address this challenge, there is a pressing need for more sophisticated modeling techniques that can accommodate both target flexibility and drug adaptability [8, 9].

Recent advancements have concentrated on the development of adaptable representations for target-drug complexes. Techniques such as Transformers are increasingly employed due to their capability to capture long-range dependencies and structural relationships, particularly in the modeling of target-drug complexes [10]. Approaches like ProteinBERT [11] and MolTrans [12] underscore the efficacy of self-attention mechanisms in encoding both target and drug structures. However, Transformer-based methods face challenges in accurately modeling local target-drug structures and capturing critical geometric and spatial relationships for the purpose of modeling natural poses.

Furthermore, through message-passing mechanisms, Graph Neural Network (GNN) effectively model both local and global dependencies, enhancing the representation of local interactions while preserving a comprehensive view of complex systems [13–15]. Recent studies have utilized GNN to address the limitations inherent in Transformers [3, 16, 17]. Nevertheless, GNN may encounter challenges in capturing long-range dependencies and lack explicit mechanisms for rotational and translational invariances, which are essential for accurately modeling the adaptive nature of target-drug binding within three-dimensional space [18–20].

To address these challenges, we propose DynaPhArM, a method specifically designed for drug-aware target-drug complex modeling. DynaPhArM introduces a specialized encoding module that captures dual-layer representations of target backbones, side chains and drugs via a cooperative scalar–vector mechanism. For target backbones, we apply embedding quantization to discretize representations into informative token spaces. These representations are then integrated using a physics-constrained interaction module based on cross-attention, modulated by atomic-level docking scores to capture structurally grounded target–drug interactions. To model the adaptive nature of binding, we employ a Diffusion Denoising Probabilistic Model (DDPM) to generate latent joint embeddings that reflect conformational flexibility. Finally, DynaPhArM integrates physical constraints through a multi-objective loss function designed to balance structure reconstruction and denoising, promoting physically realistic and informative representations of target–drug complexes. Experiments demonstrate that DynaPhArM enables more accurate 3D conformation reconstruction and improves binding affinity prediction performance.

The contributions of this paper are summarized as follows:

- We introduce DynaPhArM, a method that accurately and adaptively models the target-drug complex using an SE(3)-Transformer framework.

- DynaPhArM captures adaptive joint embeddings of the target-drug complex through the physics-constrained interaction module and diffusion module, effectively reflecting diverse conformations during the binding process.

- DynaPhArM enforces physical plausibility of 3D conformations by integrating atomic interaction constraints into the loss.

## 2 Related work

### 2.1 Flexible docking

Early key-lock theory-based docking methods initially assumed rigid target structures [21], overlooking the importance of conformational flexibility [22]. Over time, flexible docking approaches have emerged to address these limitations by incorporating the conformational flexibility of targets, drugs, or both during the docking process. As early as the 1960s, advances such as nuclear magnetic resonance (NMR) spectroscopy [23] and the first molecular dynamics (MD) simulations [24] began to reveal the inherent flexibility of proteins and its crucial role in ligand binding. These insights laid the groundwork for modern flexible docking strategies. FlexPose [20] is a graph neural network-

based model designed for target-drug structure reconstruction. It employs attention-based blocks to adaptively update features, allowing for adaptation to complex environments. PackDock [25] is a two-stage model that integrates conformation selection and induced fit mechanisms, which facilitate adaptive adjustments of side-chain conformations. FAIR [26] enhances the full-atom coordinates of both the pocket and drug while adaptively updating residue types, backbone configurations, and flexible side-chain structures within the target. EDM-Dock [27] utilizes a dynamic graph neural network to predict distance matrices and variances between target-drug nodes for generating flexible docking poses.

## 2.2 Denoising diffusion probabilistic models

DDPM [28] is a generative model that learns to produce samples by progressively reversing a diffusion process that introduces noise into data, which manages complex data distributions and generate realistic structural representations positions [29]. DiffDock-Pocket [30] employs DDPM to regenerate the translation, rotation, and torsion angles of the drug, along with the torsion angles of the side chains, effectively simulating the flexible alterations occurring during the binding process. DiffBindFR [31] conceptualizes structure reconstruction as a joint denoising process involving four variables in tangent space, thereby generating adaptive binding conformations. SurfDock [32] enhances initial random pose through denoising via DDPM, which captures complex distributions and enables the generation of adaptive drug poses within a flexible spatial framework. DynamicBind [33] utilizes DDPM to optimize conformational changes in both apo targets and drugs, modeling target flexibility through side-chain torsions and residue movements.

## 3 Method

We introduce DynaPhArM, a transformer-based diffusion model (illustrated in Figure 1) designed for drug-specific target-drug complex modeling. The model consists of three principal modules: representation encoder with a cooperative scalar–vector encoding module (3.1), adaptive joint embedding generation module (3.2) and loss function integrating physics and geometry constraints (3.3). The decoder architecture and inference procedure are provided in Appendix A and B.

### 3.1 Representation encoding module

As shown in Figure 1, DynaPhArM employs representation encoding module (target side-chain $Encoder_s$, target backbone $Encoder_b$ and drug $Encoder_d$) composed of two main components: cooperative scalar-vector representation and target embedding discretization to represent both the drug and target molecules. Both target and drug are represented as graphs. We utilize hierarchical encoding for the target by dividing it into the backbone $\mathcal{G}^b = (\mathcal{V}^b, \mathcal{E}^b)$ and the side chain $\mathcal{G}^s = (\mathcal{V}^s, \mathcal{E}^s)$. All nodes and edges are jointly parameterized by scalar and vector features (as shown in Appendix C).

**Cooperative scalar–vector representation.** To overcome the decoupling of scalar and vector channels in the original SE(3)-Transformer [34], we devise a dynamic bidirectional update mechanism (Algorithm in Appendix D) that tightly couples geometric (vector) and semantic (scalar) features. At each iteration, the passing of the node-node and edge-edge message is enhanced by gated updates: for each node $v_i$, we compute directional messages $\mathbf{h}_{ij}$ from neighbors $v_j$ that fuse its scalar descriptor, edge scalars and a learned encoding of the edge vector; Attention weights $\alpha_{ij}$ then modulate their aggregation into a residual message $\mathbf{m}_i$. The node scalar feature is updated by an multi-layer perceptron (MLP), while its vector feature undergoes an SE(3)-Equivariant correction via another MLP. Edge features are likewise refined: spherical harmonics–based geometric embeddings $\mathbf{h}_{ij}^{\text{geometric}}$ augment edge scalars through an edge-MLP, and the edge vector is shifted by a learned function of its incident node vectors.

Finally, an interaction module executes a scalar–vector inner product at each node to produce a cross-modal message $\mathbf{h}_{\text{interact}}$, which is fed back to both channels, adding squared norms of the other channel as residual boosts, to enable true bidirectional flow. We apply this cooperative encoding to the drug graph $\mathcal{G}^d$, as well as the target backbone $\mathcal{G}^b$ and side-chain $\mathcal{G}^s$ graphs. After $T$ iterations, spatially aggregated features (via global average pooling over nodes and edges) are passed through task-specific MLP to yield three embeddings: the backbone embedding $e^b$ (global fold topology), the side-chain embedding $e^s$ (local conformational variability), and the drug embedding $e^d$.

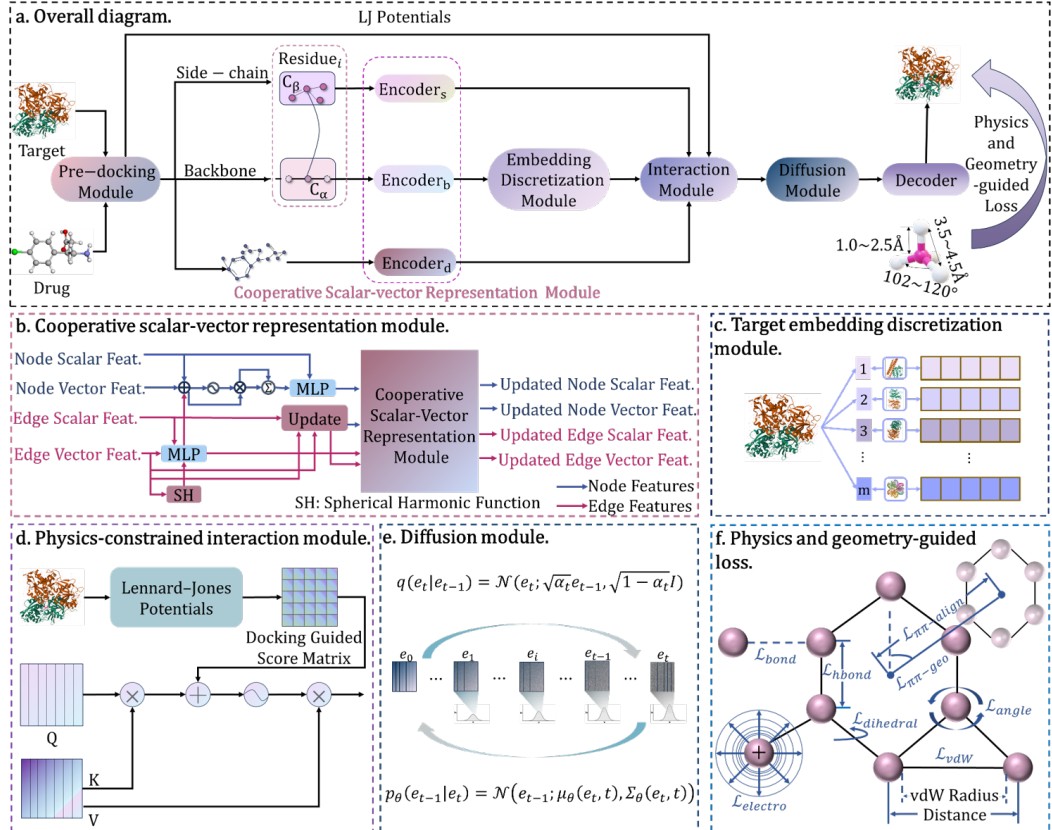

Figure 1: DynaPhArM pipeline outlines key modules for representing and modeling target-drug complexes. **a** The overall framework of DynaPhArM. The model encodes target–drug features via scalar–vector and discretization modules, integrates them with physics-aware attention, and generates 3D structures using a diffusion model under a multi-task loss enforcing structural and physical consistency. **b** The cooperative scalar-vector representation module serves to dynamically integrate geometric and semantic features of targets and drugs. **c** The discretization module facilitates the quantization of backbone embeddings into discrete tokens. **d** The physics-constrained interaction module is designed to generate embeddings for backbones, side chains, and drugs. **e** The diffusion module models adaptive complex structures based on the joint embedding derived from the physics-constrained interaction module. **f** The physics and geometry-guided loss function ensures that the predicted complex structures adhere to physically realistic interaction potentials and geometric constraints.

**Target embedding discretization.** After obtaining the enhanced embedding via scalar-vector cooperation, the target representation is discretized using the pre-trained FoldToken4 codebook [35]. This discrete encoding has been shown to more effectively capture recurring structural motifs, reduce representational variance, and minimize redundancy in the learned embeddings. Using the inherent regularity of the backbone structures compared to the conformational diversity of the side chains, we map the continuous geometric embedding $e^b$ into a discrete semantic space through vector quantization. Specifically, the high-dimensional continuous representation $e^b$ is compressed into a discrete semantic token $e^t$:

$$e^t = \text{Tokenize}(e^b) = \arg\min_{c_i \in \mathcal{C}} \|e^b - \{c_i\}_{i=1}^N\|_2 \tag{1}$$

by minimizing the Euclidean reconstruction error between the backbone embedding and the codebook vectors $\mathcal{C} = \{c_1, c_2, \ldots, c_N\}$. This discretization mechanism achieves dimension-reduced encoding of 3D structures while preserving backbone topological features, significantly enhancing storage efficiency and computational tractability of target representations.

## 3.2 Adaptive joint embedding generation

In this section, we adopt the physics-constrained interaction module and diffusion module to generate adaptive joint embedding for the target-drug complex.

**Physics-constrained interaction module.**   To capture the adaptive nature of target–drug interactions in a drug-specific manner, we incorporate docking-derived interaction priors into the attention computation. Specifically, we extend the conventional cross-attention formulation by integrating a physics-constrained interaction matrix that reflects atomic-level docking scores, as shown in Equation 2:

$$\text{head}_i = \text{Softmax}\left(\frac{Q_i K_i^\top + \gamma_i S_{ij}^{xy}}{\sqrt{d_k}}\right) V_i \tag{2}$$

Here, $Q_i = e^x W_i^Q$, $K_i = e^y W_i^K$, and $V_i = e^z W_i^V$, where $x, y, z \in \{b, s, d\}$ denote the backbone ($b$), side-chain ($s$), and drug ($d$) embeddings respectively, depending on the directional pair under consideration. Let $e^b \in \mathbb{R}^{L_b \times d_e}$, $e^s \in \mathbb{R}^{L_s \times d_e}$, and $e^d \in \mathbb{R}^{L_d \times d_e}$ be the corresponding input features, with $W_i^Q, W_i^K, W_i^V \in \mathbb{R}^{d_e \times d_k}$ denoting learnable projection matrices. $L_b$ represents the number of nodes in the protein backbone. Each node in the backbone graph corresponds to a single residue. $L_s$ represents the number of nodes in the protein side-chain. In our atom-level graph for the side-chain, each node corresponds to a single heavy atom. $d_e$ represents the dimensionality of the node feature embeddings. It is a hyperparameter that defines the length of the feature vector used to represent each node, whether it is a residue in the backbone or an atom in the side-chains or the drug. The learnable scalar $\gamma_i$ controls the influence of the physical prior on the final attention distribution.

The interaction matrix $S_{ij}^{xy}$ is computed by using Lennard–Jones (LJ) potentials of target–drug conformation modeled by AutoDock tool. For each directional pair $x, y \in \{b, s, d\}$, we define

$$S_{ij}^{xy} = 4\varepsilon_{ij}^{xy}\left[\left(\frac{\sigma_{ij}^{xy}}{r_{ij}}\right)^{12} - \left(\frac{\sigma_{ij}^{xy}}{r_{ij}}\right)^{6}\right] \tag{3}$$

where $r_{ij} = \|p_i - p_j\|$ is the Euclidean distance. $S_{ij}^{xy}$ serves as an interaction-aware prior that modulates the raw attention logits. By adding $\gamma_i S_{ij}^{xy}$ to $Q_i K_i^\top$, the model is guided to emphasize spatially and chemically plausible interaction sites, promoting more meaningful attention across drug–target pairs. And the LJ parameters are computed via Lorentz–Berthelot rules:

$$\sigma_{ij} = (\sigma_i + \sigma_j)/2, \quad \varepsilon_{ij} = \sqrt{\varepsilon_i \varepsilon_j}. \tag{4}$$

where $\sigma_i$ and $\sigma_j$ are the LJ size (collision-diameter) parameters of atom $i$ and atom $j$. $\varepsilon_i$ and $\varepsilon_j$ are the LJ energy-well depth parameters of atom $i$ and atom $j$.

To account for the full interaction landscape, we compute attention over six directional combinations: $b \to s$, $s \to b$, $b \to d$, $d \to b$, $s \to d$, and $d \to s$. Each head output has shape $\mathbb{R}^{L_x \times d_v}$. Concatenating all six directional outputs along the feature dimension yields a tensor of shape $\mathbb{R}^{L_x \times (6d_v)}$, which is then passed through a trainable MLP to project it back into $\mathbb{R}^{L_x \times d_e}$ to produce a compact fused embedding $e^f$. By integrating these interaction-aware priors, the model generates adaptive, drug-specific representations that reflect diverse structural or functional responses across target environments.

**Diffusion module.**   To model the adaptive nature of protein-ligand binding, we employ a conditional DDPM to generate a refined latent embedding. This module transforms an initial fused embedding, $e^f$, into a final conformation-aware embedding, $e_{refined}$, which captures structural flexibility.

The diffusion model consists of a forward process and a reverse process operating on a latent variable $z$. The forward process, $q$, progressively adds Gaussian noise to the ground-truth embedding, $e_{gt}$, over $T$ timesteps. We define the starting point of this process as $z_0 = e_{gt}$:

$$q(z_{1:T}|z_0) = \prod_{t=1}^{T} q(z_t|z_{t-1}), \quad \text{where } q(z_t|z_{t-1}) = \mathcal{N}(z_t; \sqrt{1-\beta_t} z_{t-1}, \beta_t \mathbf{I}) \tag{5}$$

The reverse process is modeled by a neural network $p_\theta$ that learns to iteratively denoise a noisy variable $z_t$, critically conditioned on the fused embedding $e^f$:

$$p_\theta(z_{0:T}|e^f) = p(z_T)\prod_{t=1}^{T} p_\theta(z_{t-1}|z_t, e^f), \quad \text{where } p(z_T) = \mathcal{N}(z_T; 0, \mathbf{I}) \tag{6}$$

During inference, the refined embedding $e_{refined}$ is generated by starting from a noise vector $z_T \sim \mathcal{N}(0, \mathbf{I})$ and iteratively applying the learned denoising steps $p_\theta$ for $t = T, \dots, 1$. The final output of this process is $e_{refined} = z_0$. Conditioning on $e^f$ at each step ensures that the resulting embedding accurately reflects the interaction patterns specific to the protein-ligand pair.

## 3.3 Physics and geometry-guided loss design

To ensure the biophysical plausibility of the generated target-drug conformations, we integrate fundamental physics and geometric information as constraints to loss function $L_{phy-geo}$. The feasibility of generated structures is enforced through seven geometrically regularization terms: bond lengths, bond angles, dihedral angles, vdW radius, electrostatic interaction, hydrogen bonding interaction and $\pi - \pi$ interaction, details of which can be found in Appendix E. Then, the physics and geometry-guided loss is the sum of the above seven items:

$$\mathcal{L}_{phy-geo} = \mathcal{L}_{bond} + \mathcal{L}_{angle} + \mathcal{L}_{\text{dihedral}} + \mathcal{L}_{vdW} + \mathcal{L}_{\text{electro}} + \mathcal{L}_{\text{hbond}} + \mathcal{L}_{\pi-\pi} \tag{7}$$

In addition, we consider the objective of complex modeling task, which focuses on modeling the all-atom coordinates $\hat{r}_i$ of the predicted complex derived from the decoder relative to their ground-truth positions $r_i$, which is defined as:

$$\mathcal{L}_{\text{structure}} = \frac{1}{M}\sum_{i=1}^{M} \| \hat{r}_i - r_i \|^2 \tag{8}$$

where $M$ represents the number of atomic coordinates in the complex.

Within the embedding space, the noise prediction task is critical for the diffusion-based denoising process, aiming to minimize the error between predicted $\hat{\epsilon}_\theta$ and true $\epsilon$ noise, conditioned on the corrupted embedding $e_t$ and the time step t:

$$\mathcal{L}_{\text{noise}} = \mathbb{E}_{t,\epsilon}[\|\epsilon - \hat{\epsilon}_\theta(\mathrm{e}_t, t)\|^2] \tag{9}$$

The total loss is defined as a weighted sum of three objectives:

$$\mathcal{L}_{\text{total}} = \alpha_1 \mathcal{L}_{\text{noise}} + \alpha_2 \mathcal{L}_{\text{structure}} + \alpha_3 \mathcal{L}_{\text{phy-geo}} \tag{10}$$

where $\alpha_1$, $\alpha_2$, and $\alpha_3$ are dynamically adjusted weights that balance the contributions of each task during training. The detailed adaptive weight adjustment method can be found in Appendix F.

## 4 Experiments and results

In this section, we comprehensively evaluated DynaPhArM's performance across multiple benchmarks (Section 4.1) in complex structure modeling , binding affinity prediction, cross-docking assessment (Section 4.2) and conformational ensemble docking performance in Appendix G. We also did a few ablation studies to show the functions of main modules in Section 4.3. Then, we conducted the case study to validate the drug-specific binding in Section 4.4 and its ability to capture target adaptive changes in the drug binding process in Appendix H. We finally analyzed the model's sensitivity to the physical priors (Appendix I).

## 4.1 Experimental setup

**Dataset.** We curated a comprehensive dataset of 21,762 high-quality target-drug complexes from the Protein Data Bank (PDB) and 2,417 validated drugs from DrugBank. Each drug was standardized and structurally validated. To prevent data leakage and ensure generalizability, complexes were grouped by 30% sequence identity, with entire clusters assigned to either the training (80%) or validation (20%) sets, guaranteeing no sequence overlap between subsets. The details of the curated data and other datasets we used can be found in Appendix J.

**Baselines.** We evaluated our model against GNN-based methods (FlexPose [20], TankBind [36]), diffusion-based models (PackDock [25], DynamicBind [33], DiffDock [2], DiffBindFR [31], DiffDock-Pocket [30], SurfDock [32]) and conventional docking tools (Glide [37], RosettaLigand [38] and AutoDock [39]).

**Evaluation metrics.** The proposed method evaluates both structural and functional performance. For structural accuracy, we compute the RMSD of the entire target-drug complex (overall RMSD), the ligand's 3D coordinates (L-RMSD), and the side-chain conformations (sc-RMSD). Corresponding success rates are reported as the percentage of cases with RMSD values less than 2 Å, including overall success rate, ligand success rate, and side-chain success rate, which respectively reflect the reconstruction fidelity of the full complex, the drug molecule, and side-chain positioning. For functional performance, we assess binding affinity prediction accuracy using the Pearson correlation coefficient (PCC), root mean square error (RMSE), Spearman correlation coefficient (Spearman) and mean absolute error (MAE) between predicted and experimental affinities.

## 4.2 Main results

**Complex structure modeling.** We evaluated the performance of DynaPhArM on complex structure modeling and compared it with a range of baseline methods as shown in Table 1. Among all the methods, DynaPhArM achieved the best overall performance, with the lowest overall RMSD, sc-RMSD and L-RMSD. Moreover, it demonstrated the highest success rates, achieving 61.60% for L-RMSD <2 Å, 70.50% for sc-RMSD <2 Å and 65.30% for overall RMSD <2 Å, which highlight DynaPhArM's ability to effectively model both global and local structural details of target-drug complexes. These results demonstrate that DynaPhArM provides significant advantages over existing methods in modeling accurate and physically plausible target-drug complexes, particularly in cases requiring adaptive flexibility. The running time in reconstructing task of all of these methods can be found in Appendix K.

Table 1: Performance of different methods in reconstructing complex structures within the test set, evaluated by RMSD (Å) and success rate.

| Method | RMSD (↓) | | | Success Rate (↑) | | |
|---|---|---|---|---|---|---|
| | Overall | Side-chain | Ligand | Overall RMSD < 2 Å | sc-RMSD < 2 Å | L-RMSD < 2 Å |
| Glide [37] | 11.20±4.92 | 9.37±4.32 | 15.01±4.98 | 18.79% | 19.73% | 10.31% |
| RosettaLigand [38] | 9.83±4.01 | 9.19±4.13 | 19.32±4.84 | 20.97% | 21.02% | 8.33% |
| AutoDock [39] | 8.60±4.73 | 7.05±4.01 | 13.29±4.56 | 21.06% | 22.80% | 12.56% |
| TankBind [36] | 5.36±3.14 | 4.48±3.25 | 14.66±5.08 | 23.51% | 29.10% | 11.34% |
| DiffDock [2] | 3.77±2.85 | 2.53±1.40 | 9.73±4.10 | 35.71% | 57.50% | 18.00% |
| DiffDock-Pocket [30] | 3.19±1.40 | 1.27±1.56 | 6.24±4.22 | 39.10% | 67.20% | 25.65% |
| DynamicBind [33] | 3.01±1.88 | 1.36±0.67 | 3.25±2.67 | 38.72% | 70.10% | 51.77% |
| DiffBindFR [31] | 2.98±1.38 | 0.61±0.38 | 2.73±1.30 | 50.98% | 55.30% | 57.22% |
| SurfDock [32] | 2.55±2.35 | 0.55±1.37 | 4.59±2.46 | 58.23% | 70.00% | 46.16% |
| PackDock [25] | 2.52±1.44 | 0.52±0.37 | 2.80±1.42 | 56.54% | 68.10% | 57.89% |
| FlexPose [20] | 2.44±2.83 | 0.44±0.34 | 3.48±2.20 | 60.17% | 70.20% | 30.24% |
| **DynaPhArM** | **2.01±1.85** | **0.29±1.04** | **2.17±1.45** | **65.30%** | **70.50%** | **61.60%** |

(↑) / (↓) denotes higher / lower is better; Top 1 and Top 2 are highlighted with **bold** and underlined, respectively.

**Cross-docking assessment.** Cross-docking serves as a stringent, application-oriented benchmark by placing ligands into non-cognate receptor conformations, thereby probing a docking model's ability to generalize beyond crystallographically aligned complexes. In the apo–holo scenario (CDK2, EGFR, CASF2016-Drug), ligands must navigate collapsed or misaligned binding sites in ligand-free receptors, demanding that the model infer plausible induced-fit rearrangements of side chains and backbone to reconstruct binding-competent geometries. The holo–holo setting (DUDE27-HoloEns) removes the apo bias but introduces geometric mismatch among pockets induced by different co-crystallized ligands, so success hinges on recognizing invariant physicochemical interaction patterns across diverse induced-fit landscapes. Finally, the ANN docks into homology-modeled or unrelated receptor structures plagued by backbone shifts, rotamer errors, and absent waters or cofactors, imposing extreme structural noise. Traditional rigid-body algorithms falter under these out of distribution challenges, whereas DynaPhArM—by leveraging an SE(3)-equivariant, cooperative scalar–vector architecture and a learned joint target-drug embedding, which bridges conformational

Table 2: Performance of different methods on various cross-docking datasets, evaluated by overall success rate and overall RMSD (Å).

| Method | Overall Success Rate ↑ | | | | | Overall RMSD ↓ | | | | |
|---|---|---|---|---|---|---|---|---|---|---|
| | CDK2 | EGFR | DUDE27-HoloEns | CASF2016-Drug | ANN | CDK2 | EGFR | DUDE27-HoloEns | CASF2016-Drug | ANN |
| AutoDock [39] | 5.92% | 3.11% | 11.37% | 18.47% | 15.29% | 6.12 | 6.37 | 6.04 | 4.37 | 5.37 |
| Glide [37] | 15.17% | 9.38% | 13.10% | 16.32% | 13.38% | 5.21 | 6.87 | 5.72 | 4.94 | 4.86 |
| DiffDock [2] | 29.83% | 33.55% | 34.78% | 32.72% | 21.40% | 4.06 | 5.46 | 3.17 | 3.46 | 4.02 |
| TankBind [36] | 45.20% | 41.83% | 31.69% | 20.98% | 24.01% | 2.17 | 2.21 | 3.14 | 4.10 | 3.90 |
| DiffBindFR [31] | 39.45% | 41.05% | 34.24% | 63.02% | 54.56% | 1.85 | 2.58 | 3.28 | 2.23 | 2.43 |
| DiffDock-Pocket [30] | 44.28% | 30.75% | 29.05% | 40.81% | 30.12% | 1.82 | 5.50 | 3.91 | 3.12 | 3.11 |
| PackDock [25] | 50.64% | 28.37% | 20.46% | 61.13% | 42.63% | 1.73 | 4.02 | 4.56 | 2.31 | 2.94 |
| FlexPose [20] | 48.56% | 33.22% | 26.81% | 69.24% | 36.74% | 1.80 | 3.13 | 5.13 | 1.38 | 3.05 |
| **DynaPhArM** | **56.01%** | **43.46%** | **40.52%** | **70.06%** | **60.85%** | **1.75** | **2.14** | **2.80** | **1.05** | **2.28** |

(↑) / (↓) denote that higher/lower is better. Top 1 and Top 2 results are highlighted with **bold** and underlined, respectively.

Table 3: Performance of different methods in predicting target-drug binding affinity, evaluated by PCC, RMSE, Spearman, and MAE.

| Method | PCC ↑ | RMSE ↓ | Spearman ↑ | MAE ↓ |
|---|---|---|---|---|
| $K_{DEEP}$ [40] | 0.734 | 1.50 | 0.725 | 1.12 |
| ECIFGraph::HM-Holo-Apo [41] | 0.781 | 1.42 | 0.765 | 1.08 |
| FlexPose [20] | 0.793 | 1.39 | 0.780 | 1.06 |
| Interformer [42] | 0.802 | 1.27 | 0.790 | 1.04 |
| ELGN [43] | 0.805 | 1.31 | 0.795 | 1.07 |
| GIGN [44] | 0.807 | 1.34 | 0.800 | 1.10 |
| GraphWater-Net [45] | 0.814 | 1.28 | 0.810 | 1.05 |
| **DynaPhArM** | **0.820** | **1.20** | **0.815** | **1.00** |

(↑) / (↓) denotes a higher / lower number is better; Top 1 and Top 2 results are highlighted with **bold** and underlined, respectively.

gaps, disentangles ligand- and receptor-specific features, and maintains high accuracy and low RMSD even in the most distorted scenarios.

**Binding affinity prediction.** We evaluated DynaPhArM's performance in the downstream binding affinity prediction task. $K_{DEEP}$ [40] is based on the Convolutional Neural Network (CNN). ECIFGraph::HM-Holo-Apo [41] and GraphWater-Net [45] are GNN-based methods. FlexPose [20] and Interformer [42] are transformer-based methods. ELGN [43] and GIGN [44] are EGNN-based methods. As shown in Table 3, DynaPhArM achieved the highest PCC of 0.820 and the lowest RMSE of 1.20, outperforming the second best method by 0.006 and 0.07. This superior performance validates DynaPhArM's efficacy not only in modeling and structure prediction but also as a robust tool for affinity prediction, making it a versatile and powerful model for structure-based drug design.

## 4.3 Ablation experiments

To evaluate the contributions of key components in our model, we conducted ablation experiments by systematically removing pre-docking strategy, physical features, cooperative scalar-vector representation module, target embedding discretization module, physics-constrained interaction module, diffusion module and physical constraints, as illustrated in Table 4.

**Criticality of the pre-docking strategy.** The pre-docking strategy initializes drug poses to avoid local energy minima. Adding a pre-docking strategy modestly improves the drug success rate by 3.14%, demonstrating that accurate geometric initialization acts synergistically with structural modeling to preserve global conformational fidelity, preventing pose divergence during refinement.

**Importance of physical features.** Physical features (e.g., atomic mass, charge) offer intrinsic molecular descriptors. Removing them caused a notable rise in sc-RMSD from 0.29Å to 0.31Å and a decrease in drug success rate by 1.04%, which highlights that explicit physical priors enhance local structural alignment and enhance atomic-level discriminability.

**Effectiveness of the cooperative scalar-vector representation module.** The cooperative scalar-vector representation module is designed to model the structural interdependencies. Removing the cooperative scalar-vector representation module significantly degraded side-chain prediction accuracy, increasing RMSD from 0.29Å to 2.04Å, while the drug success rate dropped by 17.47%, highlighting the critical importance of this module in maintaining accurate structural predictions.

**Role of the target embedding discretization module.** Discretization encodes local geometric details at fine resolution. Its removal resulted in a substantial degradation: the overall RMSD increased from 2.01Å to 3.34Å, and the success rate dropped from 61.60% to 41.34%. These results emphasize that discretized spatial encoding is indispensable for capturing spatial granularity critical for precise structure generation.

**Effect of the physics-constrained interaction module.** Physics-constrained interactions impose inter-molecular force priors. Eliminating this component increased the overall RMSD from 2.01Å to 2.11Å and reduced drug success rate by 8.69%. This suggests that energy-aware modeling facilitates spatial compatibility between target and drug, contributing to biologically valid predictions.

**Contribution of the diffusion module.** Diffusion modeling introduces denoising dynamics to capture conformation flexibility. Removing it led to severe performance loss, with overall RMSD increasing by 2.98Å and drug success rate dropping by 30.79%. This underlines that the generative denoising process is fundamental for modeling complex structural distributions in a physically consistent manner.

**Necessity of physical constraints.** Physical constraints include seven physical and geometrical terms to ensure structural plausibility. Adding these constraints significantly mitigates deviations, the overall RMSD decreases by 0.79Å, and overall success rate rises by 14.76%, which demonstrates that physical constraints prioritize local atomic realism over global sampling, leveraging physicochemical principles to refine modeling.

Table 4: Experimental results on ablation study, which summarizes the performance of models with various components (✓ / ✗), highlighting the impact of each element. a: pre-docking strategy, b: physical features, c: cooperative scalar-vector representation module, d: target embedding discretization module, e: physics-based interaction module, f: diffusion module, g: physical constraints.

| Components | | | | | | | RMSD ↓ | | | Success Rate ↑ | | |
|---|---|---|---|---|---|---|---|---|---|---|---|---|
| a | b | c | d | e | f | g | Overall | Side-chain | Ligand | Overall RMSD <2Å | sc-RMSD <2Å | L-RMSD <2Å |
| ✓ | ✓ | ✓ | ✓ | ✓ | ✓ | ✓ | **2.01 Å** | **0.29 Å** | **2.17 Å** | **65.30%** | **70.50%** | **61.60%** |
| ✗ | ✓ | ✓ | ✓ | ✓ | ✓ | ✓ | 2.07 Å | 0.48 Å | 2.31 Å | 64.17% | 69.17% | 58.46% |
| ✓ | ✗ | ✓ | ✓ | ✓ | ✓ | ✓ | 2.05 Å | 0.31 Å | 2.73 Å | 64.66% | 70.15% | 60.56% |
| ✓ | ✓ | ✗ | ✓ | ✓ | ✓ | ✓ | 3.22 Å | 2.04 Å | 4.96 Å | 49.72% | 62.01% | 44.13% |
| ✓ | ✓ | ✓ | ✗ | ✓ | ✓ | ✓ | 3.34 Å | 1.89 Å | 5.13 Å | 47.01% | 67.88% | 41.34% |
| ✓ | ✓ | ✓ | ✓ | ✗ | ✓ | ✓ | 2.11 Å | 0.59 Å | 2.70 Å | 60.87% | 57.45% | 52.91% |
| ✓ | ✓ | ✓ | ✓ | ✓ | ✗ | ✓ | 4.99 Å | 3.72 Å | 4.08 Å | 40.45% | 36.67% | 30.81% |
| ✓ | ✓ | ✓ | ✓ | ✓ | ✓ | ✗ | 2.80 Å | 1.93 Å | 5.42 Å | 50.54% | 66.21% | 43.02% |
| ✗ | ✓ | ✗ | ✓ | ✓ | ✓ | ✓ | 4.53 Å | 3.65 Å | 3.81 Å | 41.08% | 38.90% | 35.27% |
| ✓ | ✓ | ✗ | ✓ | ✓ | ✓ | ✗ | 4.60 Å | 3.06 Å | 4.26 Å | 37.19% | 37.12% | 30.78% |
| ✗ | ✓ | ✗ | ✓ | ✓ | ✓ | ✗ | 5.01 Å | 5.17 Å | 6.11 Å | 29.23% | 25.45% | 26.09% |

(↑) / (↓) denotes higher / lower is better. Top 1 and Top 2 results are **bold** and underlined.

## 4.4 Case study

We evaluate DynaPhArM through two case studies: drug-specific binding conformations on a shared target (3APW) using two drugs and reconstruction accuracy compared to baselines (Appendix H) .

Disopyramide is an antiarrhythmic agent used to treat ventricular arrhythmias by stabilizing cardiac electrical activity [46] and 3APW is an experimentally validated target for this drug. As illustrated in Figure 2, for Disopyramide binding to the target, DynaPhArM achieved high structural accuracy, with an overall RMSD of 0.92 Å, which underscores its capacity to reconstruct the precise spatial organization of the complex, including critical molecular interactions governing therapeutic action.

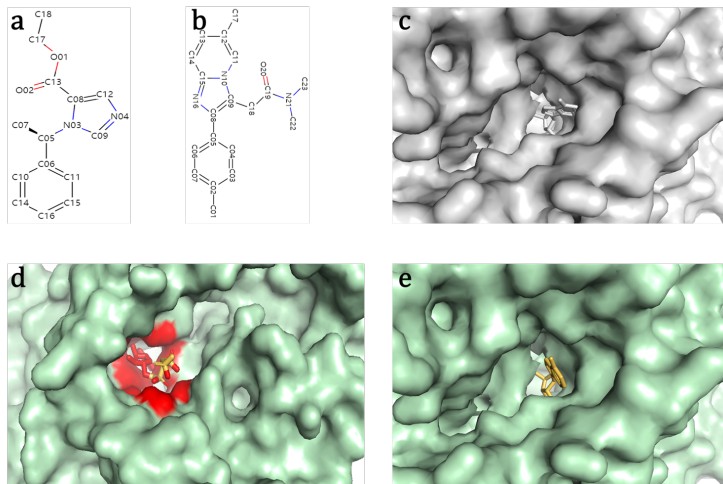

Figure 2: The case of DynaPhArM on one target 3APW and different drugs. **a** The 2D representation of the drug molecule Disopyramide (DP0). **b** The 2D representation of the drug molecule Zolpidem (R5R). **c, e** The experimental in grey and predicted (green target surface and yellow DP0 sticks) structures of complex forming by DP0 and 3APW, respectively. **d** The predicted structure of (green target surface, yellow R5R sticks and red collision region) complex forming by R5R and 3APW.

In contrast, Zolpidem is a sedative-hypnotic drug used for short-term management of insomnia by modulating GABA-A receptor activity [47], which is not naturally binded to 3APW. DynaPhArM accurately predicted structural rearrangements, including side-chain closure around the ligand, which is a hallmark of specific engagement, demonstrating both structural fidelity and adaptive sensitivity.

## 5   Conclusion

In this study, we present DynaPhArM, a framework based on SE(3)-Equivariant Transformer designed to simulate the flexible and adaptive nature of target-drug interactions. DynaPhArM utilizes a cooperative scalar-vector representation approach and discretazation mechanism that facilitates efficient and scalable modeling. By integrating adaptive embeddings with a physics-constrained MTL framework, DynaPhArM guarantees the generation of chemically accurate and physically plausible 3D structural models of target-drug complexes. DynaPhArM provides deeper insights into target-drug interactions, enabling the development of more effective and individualized therapeutic strategies. Extending DynaPhArM to complex, multi-target systems could boost discovery efficiency and accuracy but risks overfitting, higher validation costs, or biosecurity.

## Acknowledgements

This work was supported by the National Natural Science Foundation of China (62322215, 62532017, 62402488, 62372279), the Hunan Provincial Natural Science Foundation of China (2025JJ60387), and the Natural Science Foundation of Shandong Province (ZR2025QB62, ZR2023MF119). This study was also supported in part by the High-Performance Computing Center of Central South University.

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

# A Decoder architecture

The decoder $f_{\text{dec}}$ is responsible for transforming the final diffusion-denoised fused representation $e_f^0 \in \mathbb{R}^{N \times d}$—which jointly encodes physics-aware target-drug interaction semantics—into a physically plausible 3D structure $\hat{C} = \{\hat{r}_i \in \mathbb{R}^3\}_{i=1}^N$, where $N$ denotes the number of nodes (residues and atoms).

The decoder is implemented as an $L$-layer SE(3)-Equivariant graph transformer, where each node $i$ is initialized with the latent feature $h_i^{(0)} = e_f^{0,i}$ from diffusion and an initial coordinate $r_i^{(0)}$ obtained via a pre-docking strategy:

$$h_i^{(0)} = W^{(0)} e_f^{0,i} + b^{(0)}, \quad r_i^{(0)} = r_i^{\text{init}} \tag{11}$$

At each layer $\ell = 1, \ldots, L$, the decoder performs the following operations:

## A.1 Equivariant attention

At each layer $\ell$, the decoder refines node-level representations through a self-attention mechanism that integrates contextual information across the target–drug interaction embedding. Specifically, each node $i$ with scalar feature $h_i^{(\ell-1)} \in \mathbb{R}^d$ computes attention weights over its neighbors by projecting its embedding into a query vector:

$$Q_i^{(\ell)} = h_i^{(\ell-1)} W_Q^{(\ell)}, \tag{12}$$

where $W_Q^{(\ell)} \in \mathbb{R}^{d \times d}$ is a learnable projection matrix at layer $\ell$.

Similarly, all other nodes $j$ are projected into key and value vectors:

$$K_j^{(\ell)} = h_j^{(\ell-1)} W_K^{(\ell)}, \quad V_j^{(\ell)} = h_j^{(\ell-1)} W_V^{(\ell)} \tag{13}$$

where $W_K^{(\ell)}, W_V^{(\ell)} \in \mathbb{R}^{d \times d}$ are the key and value projection matrices, respectively.

The attention weight between node $i$ and node $j$ is computed via scaled dot-product attention:

$$a_{ij}^{(\ell)} = \frac{\exp\left(Q_i^{(\ell)} \cdot K_j^{(\ell)} / \sqrt{d}\right)}{\sum_{k=1}^N \exp\left(Q_i^{(\ell)} \cdot K_k^{(\ell)} / \sqrt{d}\right)}, \tag{14}$$

where $d$ is the hidden dimension and $a_{ij}^{(\ell)}$ defines the normalized attention score assigned to node $j$ when updating node $i$.

Using these weights, node $i$ aggregates contextual information from all neighbors:

$$\tilde{h}_i^{(\ell)} = \sum_{j=1}^N a_{ij}^{(\ell)} V_j^{(\ell)}, \tag{15}$$

where $\tilde{h}_i^{(\ell)}$ represents the aggregated semantic message from the neighborhood of $i$.

Finally, the new node representation is updated using a residual connection followed by layer normalization:

$$h_i^{(\ell)} = \text{LayerNorm}\left(h_i^{(\ell-1)} + \tilde{h}_i^{(\ell)}\right), \tag{16}$$

where $h_i^{(\ell)}$ is the refined embedding at layer $\ell$.

## A.2 Message passing

To incorporate distance-aware relational information into node-level reasoning, we employ a message passing mechanism that explicitly encodes pairwise Euclidean distances into high-dimensional edge features. At each layer $\ell$, for each node pair $(i, j)$, we first compute their spatial distance based on current coordinates:

$$d_{ij}^{(\ell-1)} = \|r_i^{(\ell-1)} - r_j^{(\ell-1)}\|_2, \tag{17}$$

where $r_i^{(\ell-1)} \in \mathbb{R}^3$ is the coordinate of node $i$ at layer $\ell - 1$, and $d_{ij}^{(\ell-1)}$ denotes the Euclidean distance between nodes $i$ and $j$.

The scalar distance $d_{ij}^{(\ell-1)}$ is then passed through a radial basis function (RBF) expansion to obtain a smooth, high-dimensional representation:

$$\phi_k(d_{ij}) = \exp\left(-\beta_k(d_{ij} - \mu_k)^2\right), \quad k = 1, \ldots, K, \tag{18}$$

where $\mu_k$ and $\beta_k$ are the center and bandwidth of the $k$-th Gaussian kernel, respectively. The output $\{\phi_k(d_{ij})\}_{k=1}^K$ is a $K$-dimensional vector encoding the relative spatial proximity between nodes $i$ and $j$.

Finally, we compute the edge-wise message $m_{ij}^{(\ell)}$ by integrating both node-level semantic information and the geometric encoding:

$$m_{ij}^{(\ell)} = \phi_m\left(h_i^{(\ell)}, \ h_j^{(\ell)}, \ \{\phi_k(d_{ij})\}_{k=1}^K\right), \tag{19}$$

where $h_i^{(\ell)} \in \mathbb{R}^d$ and $h_j^{(\ell)} \in \mathbb{R}^d$ are the current scalar embeddings of nodes $i$ and $j$, and $\phi_m(\cdot)$ denotes a learnable MLP that fuses semantic and geometric inputs into a unified message representation.

### A.3 Coordinate update

To refine the atomic coordinates in an SE(3)-Equivariant manner, we apply a direction-aware coordinate update based on the message $m_{ij}^{(\ell)}$ computed in A.2. For each node pair $(i, j)$, a scalar weight $w_{ij}^{(\ell)}$ is first predicted via a learnable neural function:

$$w_{ij}^{(\ell)} = \psi^{(\ell)}(m_{ij}^{(\ell)}), \tag{20}$$

where $\psi^{(\ell)}(\cdot)$ is an MLP that maps the message $m_{ij}^{(\ell)}$ into a scalar attention score that modulates the geometric influence of node $j$ on node $i$.

Next, the coordinate displacement for node $i$ is computed by aggregating normalized directional vectors weighted by $w_{ij}^{(\ell)}$:

$$\Delta r_i^{(\ell)} = \sum_j w_{ij}^{(\ell)} \cdot \frac{r_i^{(\ell-1)} - r_j^{(\ell-1)}}{d_{ij}}, \tag{21}$$

where $r_i^{(\ell-1)} \in \mathbb{R}^3$ and $r_j^{(\ell-1)} \in \mathbb{R}^3$ are the coordinates of nodes $i$ and $j$ at layer $\ell - 1$, and $d_{ij}$ is their Euclidean distance as previously defined. The term $(r_i - r_j)/d_{ij}$ is the unit direction vector pointing from $j$ to $i$, ensuring the update direction is geometrically meaningful and rotation-equivariant.

Finally, the coordinates of node $i$ are updated via residual addition:

$$r_i^{(\ell)} = r_i^{(\ell-1)} + \Delta r_i^{(\ell)}, \tag{22}$$

producing refined coordinates $r_i^{(\ell)}$ at layer $\ell$. The additive update formulation allows gradual correction of initial positions while preserving SE(3) equivariance, as the entire update depends only on relative positions and scalar weights.

### A.4 Final prediction

After $L$ layers of equivariant updates, each node $i$ is associated with a refined coordinate $r_i^{(L)} \in \mathbb{R}^3$. To ensure that the final predicted structure is invariant to global translations, we subtract the centroid of all node positions before output:

$$\hat{r}_i = r_i^{(L)} - \frac{1}{N} \sum_{j=1}^N r_j^{(L)}, \qquad \hat{C} = \{\hat{r}_i\}_{i=1}^N \tag{23}$$

where $N$ denotes the number of nodes, and $\hat{r}_i \in \mathbb{R}^3$ is the final translation-normalized coordinate of node $i$. The term $\frac{1}{N}\sum_{j=1}^{N} r_j^{(L)}$ represents the centroid of the entire structure at the final layer, and its subtraction centers the predicted coordinates around the origin.

This normalization step guarantees that the predicted structure $\hat{C}$ remains invariant under global translations of the input coordinate frame, preserving the SE(3)-Equivariant property of the overall architecture.

### A.5 Multi-Objective loss and physical geometry constraints

The coordinate predictions $\hat{C}$ are supervised under a multi-objective loss function that enforces physical plausibility, which can be found in 3.3.

## B Inference algorithm

---

**Algorithm 1** Structure Refinement via Diffusion and Geometric Constraints

---

1: **Input:** Target structure $P$, drug SMILES $S$
2: **Output:** Predicted binding pose $\hat{C}$ (3D coordinates)
3: Convert $S$ to 3D drug structure $D$ via RDKit
4: Generate initial pose $C_0$ by docking $P$ and $D$ with AutoDock
5: Build backbone graph $\mathcal{G}^b$, side-chain graph $\mathcal{G}^s$ from $P$
6: Build drug graph $\mathcal{G}^D$ from $D$
7: $e^b \leftarrow \text{Encoder}_{\text{backbone}}(\mathcal{G}^b)$
8: $e^{\text{d}} \leftarrow \text{Encoder}_{\text{drug}}(\mathcal{G}^D)$
9: $e^s \leftarrow \text{Encoder}_{\text{side-chain}}(\mathcal{G}^s)$
10: $e^{\text{t}} \leftarrow \text{Tokenize}(e^b)$
11: $e^f \leftarrow \text{PhysicsAttention}(e^{\text{t}}, e^{\text{d}})$
12: $e_f \leftarrow \text{Linear}(e^f)$
13: Sample a pure noise vector: $z_T \sim \mathcal{N}(0, \mathbf{I})$
14: **for** $t = T$ down to 1 **do**
15: $\quad \hat{\boldsymbol{\epsilon}}_\theta \leftarrow \text{DenoisingNet}_\theta(z_t, t, e_f)$ {Conditioned on $e_f$}
16: $\quad z_{t-1} \leftarrow \frac{1}{\sqrt{\alpha_t}}\left(z_t - \frac{1-\alpha_t}{\sqrt{1-\bar{\alpha}_t}}\hat{\boldsymbol{\epsilon}}_\theta\right)$
17: **end for**
18: $e_{refined} \leftarrow z_0$
19: $\hat{C} \leftarrow \text{Decoder}(e_{refined})$

---

## C Graph feature construction

Both target and drug are represented as graphs. We utilize hierarchical encoding for the target by dividing it into the backbone $\mathcal{G}^b = (\mathcal{V}^b, \mathcal{E}^b)$ and the side chain $\mathcal{G}^s = (\mathcal{V}^s, \mathcal{E}^s)$. All nodes and edges are jointly parameterized by scalar and vector features. For target, the backbone graph encodes the global structural scaffold, preserving its rigid tertiary folding through continuous peptide bond connectivity. For each node, scalar features include amino acid type, residue ID from the sequence, hydrophobicity and secondary structure, while vector features incorporate geometric information including Cartesian coordinates of the $C_\alpha$ atom, two dihedral angles and four direction vectors. Foe each edge, scalar edge features include Euclidean distance between two adjacent residues and bond type. Vector edge features include the direction vector of two adjacent residues. Meanwhile, the side-chain graph $\mathcal{G}^s = (\mathcal{V}^s, \mathcal{E}^s)$ of target captures localized chemical environments at binding sites, representing rotatable bonds and functional group orientations critical for molecular recognition. Scalar features include atom type, atom number, B-factor, functional groups type, implicit valence electrons, aromaticity, electrostatic charge and hybridization state, while vector features include the atom coordinates and five dihedral angles, which are represented by their sine and cosine values. Scalar edge features are same as the backbone. The atomic graph $\mathcal{G}^s = (\mathcal{V}^s, \mathcal{E}^s)$ explicitly models conformationally flexible side-chain atoms (e.g., $\chi$ -angle rotations in tyrosine residues), focusing

on their spatial rearrangements during drug binding [31]. Additionally, the atomic graph $\mathcal{G}^d$ is also represented by physical and chemical properties and geometric features, such as element type, partial charge, hybridization state, chirality, aromaticity, atomic mass.

## D  Cooperative scalar-vector representation algorithm

---
**Algorithm 2** Scalar-Vector Feature Interaction with SE(3)-Equivariant Update

---
1: **Input:**
2:     Node scalar features $\mathbf{f}_{\text{scalar}}(v_i)$, vector features $\mathbf{f}_{\text{vector}}(v_i)$ for all $v_i \in \mathcal{V}$
3:     Edge scalar features $\mathbf{f}_{\text{scalar}}(e_{ij})$, vector features $\mathbf{f}_{\text{vector}}(e_{ij})$ for all $e_{ij} \in \mathcal{E}$
4: **Output:** Updated $\mathbf{f}_{\text{scalar}}(v_i)$, $\mathbf{f}_{\text{vector}}(v_i)$, $\mathbf{f}_{\text{scalar}}(e_{ij})$, $\mathbf{f}_{\text{vector}}(e_{ij})$
5: Initialize iteration $t \leftarrow 0$, maximum iterations $T$
6: **repeat**
7:     **for** each node $v_i \in \mathcal{V}$ **do**
8:         **for** each neighbor $v_j \in \mathcal{N}(v_i)$ **do**
9:             $\mathbf{r}_{ij} \leftarrow \mathbf{f}_{\text{vector}}(v_j) - \mathbf{f}_{\text{vector}}(v_i)$
10:             $\theta_{ij}, \phi_{ij} \leftarrow \text{SphericalCoordinates}(\mathbf{r}_{ij})$
11:             $\mathbf{h}_{ij} \leftarrow \text{Concat}\Big( \mathbf{f}_{\text{scalar}}(v_i), \mathbf{f}_{\text{scalar}}(e_{ij}), \text{MLP}\big(\mathbf{f}_{\text{vector}}(e_{ij})\big) \Big)$
12:             $\alpha_{ij} \leftarrow \text{Softmax}\big(\mathbf{W}_1^\top \mathbf{h}_{ij} + \mathbf{b}_1\big)$
13:         **end for**
14:         $\mathbf{m}_i \leftarrow \sum_{j \in \mathcal{N}(v_i)} \alpha_{ij} \cdot \mathbf{h}_{ij}$
15:         $\mathbf{f}_{\text{scalar}}(v_i) \leftarrow \text{MLP}_{\text{node}}\left( \mathbf{f}_{\text{scalar}}(v_i), \mathbf{m}_i \right)$
16:         $\mathbf{f}_{\text{vector}}(v_i) \leftarrow \mathbf{f}_{\text{vector}}(v_i) + \text{MLP}_{\text{vec}}(\mathbf{m}_i)$
17:     **end for**
18:     **for** each edge $e_{ij} \in \mathcal{E}$ **do**
19:         $\mathbf{h}_{ij}^{\text{geometric}} \leftarrow \sum_{l=0}^{L} \sum_{m=-l}^{l} w_{lm} N_{lm} P_l^m(\cos\theta_{ij}) e^{im\phi_{ij}}$
20:         $\mathbf{f}_{\text{scalar}}(e_{ij}) \leftarrow \text{MLP}_{\text{edge}}\left( \mathbf{f}_{\text{scalar}}(e_{ij}), \mathbf{h}_{ij}^{\text{geometric}} \right)$
21:         $\mathbf{f}_{\text{vector}}(e_{ij}) \leftarrow \mathbf{f}_{\text{vector}}(e_{ij}) + \text{MLP}_{\text{vec-edge}}(\mathbf{f}_{\text{vector}}(v_i), \mathbf{f}_{\text{vector}}(v_j))$
22:     **end for**
23:     **Interaction Module:**
24:     **for** each node $v_i \in \mathcal{V}$ **do**
25:         $\mathbf{h}_{\text{interact}} \leftarrow \text{MLP}(\mathbf{f}_{\text{scalar}}(v_i)) \cdot \mathbf{f}_{\text{vector}}(v_i)$
26:         $\mathbf{f}_{\text{scalar}}(v_i) \leftarrow \mathbf{f}_{\text{scalar}}(v_i) + \|\mathbf{f}_{\text{vector}}(v_i)\|_2^2$
27:         $\mathbf{f}_{\text{vector}}(v_i) \leftarrow \mathbf{f}_{\text{vector}}(v_i) + \mathbf{h}_{\text{interact}} \cdot \|\mathbf{f}_{\text{scalar}}(v_i)\|_2$
28:     **end for**
29:     $t \leftarrow t + 1$
30: **until** $t \geq T$

---

## E  Detailed formulation of geometry and physical loss functions

Bond lengths penalize deviations from reference covalent bond distances (1.0–2.5 Å):

$$\mathcal{L}_{\text{bond}} = \sum_{(i,j)\in\mathcal{B}} \left( \frac{\|\mathbf{r}_i - \mathbf{r}_j\|_2 - d_{ij}^{\text{ref}}}{d_{ij}^{\text{ref}}} \right)^2 \tag{24}$$

where $\mathcal{B}$ denotes the set of bonded atom pairs, and $d_{ij}^{\text{ref}}$ are experimentally derived bond lengths.

Bond angle consistency maintains directional geometry by constraining angles (102°–120°) between consecutive bonds:

$$\mathcal{L}_{\text{angle}} = \sum_{(i,j,k)\in\mathcal{A}} \left( \cos^{-1} \left( \frac{(\mathbf{r}_i - \mathbf{r}_j) \cdot (\mathbf{r}_k - \mathbf{r}_j)}{\|\mathbf{r}_i - \mathbf{r}_j\|_2 \|\mathbf{r}_k - \mathbf{r}_j\|_2} \right) - \theta_{ijk}^{\text{ref}} \right)^2 \tag{25}$$

where $\mathcal{A}$ represents valid bond angle triplets.

To preserve the higher-order structural stability of the molecule, dihedral angle consistency constrains the torsional rotation between four sequentially bonded atoms, capturing the flexibility of the target and drug conformation:

$$\mathcal{L}_{\text{dihedral}} = \sum (i,j,k,l) \in \mathcal{D} \left( \phi_{ijkl}^{\text{pred}} - \phi_{ijkl}^{\text{true}} \right)^2 \tag{26}$$

where $\mathcal{D}$ denotes the set of dihedral angle quadruplets, $\phi_{ijkl}^{\text{pred}}$ is the predicted dihedral angle formed by atoms $i$, $j$, $k$, and $l$, and $\phi_{ijkl}^{\text{true}}$ represents the true torsional angle.

Steric exclusion enforcement prevents atomic overlaps using vdW radius thresholds:

$$\mathcal{L}_{\text{vdW}} = \sum_{i<j} \text{ReLU} \left( r_{ij}^{vdW} - \| \mathbf{r}_i - \mathbf{r}_j \|_2 \right)^2 \tag{27}$$

where $r_{ij}^{vdW}$ is the sum of atomic vdW radii.

To account for long-range Coulombic effects in the target-drug complex, we introduce a dipole–dipole interaction term based on the leading order of multipole expansion. This term enforces electrostatic plausibility by penalizing deviations from physically consistent dipole alignments:

$$\mathcal{L}_{\text{electro}} = \sum (i,j) \in \mathcal{P} \left( \frac{\mathbf{p}_i \cdot \mathbf{p}_j - 3(\mathbf{p}i \cdot \hat{\mathbf{r}}ij)(\mathbf{p}j \cdot \hat{\mathbf{r}}ij)}{|\mathbf{r}_i - \mathbf{r}_j|_2^3} \right)^2 \tag{28}$$

where $\mathcal{P}$ denotes all relevant dipole–dipole pairs, $\mathbf{p}_i$ and $\mathbf{p}_j$ are the atomic dipole moments, and $\hat{\mathbf{r}}_{ij}$ is the unit vector from atom $i$ to atom $j$.

Hydrogen bonding geometry is regularized to reflect canonical interaction patterns, enforcing appropriate donor–acceptor distances and angles between participating atoms:

$$\mathcal{L}_{\text{hbond}} = \sum_{(d,h,a)\in\mathcal{H}} \Big[ \text{ReLU}(|\mathbf{r}_d - \mathbf{r}_a| - 3.2)^2 + \text{ReLU}(2.5 - |\mathbf{r}_d - \mathbf{r}_a|)^2$$
$$+ \text{ReLU}(120 - \theta)^2 + \text{ReLU}(\theta - 180)^2 \Big] \tag{29}$$

where $\mathcal{H}$ is the set of donor–hydrogen–acceptor triplets, $r_d$ and $r_a$ are the coordinates of the donor and acceptor respectively. $\theta$ is the angle between the donor, hydrogen and acceptor atoms.

$\pi{-}\pi$ stacking geometry is regularized to promote favorable aromatic ring interactions by enforcing appropriate inter-centroid distances and planarity constraints between participating rings:

$$\mathcal{L}_{\pi\pi\text{-geo}} = \sum_{(i,j)\in\pi\pi\ \text{pairs}} \Big[ \text{ReLU}(d_{ij} - 5.0)^2 + \text{ReLU}(3.5 - d_{ij})^2 + \text{ReLU}\big(|\theta_{ij}| - 30°\big)^2 \Big]$$
$$+ \text{ReLU}\big(|\Delta x_{ij}| - 3.5\big)^2 \tag{30}$$

$$\mathcal{L}_{\pi\pi\text{-align}} = \sum_{(i,j)} \left( 1 - \frac{\mathbf{n}_i \cdot \mathbf{n}_j}{\|\mathbf{n}_i\|\|\mathbf{n}_j\|} \right)^2 \cdot \text{ReLU}(d_{ij} - 3.5) \tag{31}$$

$$\mathcal{L}_{\pi-\pi} = \mathcal{L}_{\pi\pi\text{-geo}} + \mathcal{L}_{\pi\pi\text{-align}} \tag{32}$$

where $\pi - \pi$ pairs is pairs of aromatic systems involved in $\pi - \pi$ interactions. $d_{ij}$ and $\theta_{ij}$ are the distance and angle between the centroids of the i-th and j-th $\pi - \pi$ systems respectively. $\Delta x_{ij}$ is the lateral displacement between the centers of the $\pi - \pi$ systems. $n_i$ and $n_j$ are normal vectors to the planes of the i-th and j-th $\pi - \pi$ systems.

# F Adaptive weight adjustment

In MTL frameworks, the design of loss weightings critically influences how the model balances its optimization trajectory across competing objectives. In our model, the total training loss comprises three core components: the structural loss $\mathcal{L}_{\text{structure}}$, the diffusion denoising loss $\mathcal{L}_{\text{noise}}$, and the physics and geometry-guided loss $\mathcal{L}_{\text{phy-geo}}$, jointly supervising the complex modeling.

Traditionally, scalar weights are statically assigned to each loss component. However, such static weighting cannot adapt to the non-stationary nature of training. Different tasks may converge at different rates or present varying levels of difficulty throughout training. Improper weighting may result in biased gradient updates, overfitting to easier tasks, or neglecting harder ones. To overcome this, we introduce a dynamically adaptive inverse proportional weighting strategy based on instantaneous loss magnitudes.

## F.1 Theoretical motivation

From a theoretical point of view, our strategy can be interpreted under a Min-Max loss balancing framework. Specifically, we seek to minimize the largest normalized task loss at each training step, formulated as:

$$\max_{i \in \{s,d,p\}} \left( \frac{\ell_i}{\alpha_i} \right) \tag{33}$$

where $\ell_i$ denotes the instantaneous scalar loss, $\alpha_i$ represents the corresponding adaptive weights.

Under our inverse proportional weighting rule, this ratio is approximately constant across all loss terms:

$$\frac{\ell_i}{\alpha_i} \approx \sum_j \ell_i \cdot \left( \frac{1/\ell_j}{\sum_k 1/\ell_k} \right) = \text{const.} \tag{34}$$

This normalization implies that the optimizer perceives all tasks as having equal scaled difficulty, thus promoting fairness in multi-objective convergence and avoiding task imbalance.

## F.2 Formulation of Inverse Proportional Normalization

At each training step $t$, we observe the instantaneous (i.e., per-batch) losses:

$$\mathcal{L}_{\text{structure}}^{(t)} = \ell_s, \quad \mathcal{L}_{\text{diffusion}}^{(t)} = \ell_d, \quad \mathcal{L}_{\text{phy-geo}}^{(t)} = \ell_p. \tag{35}$$

where $\ell_s, \ell_d, \ell_p \in \mathbb{R}_{>0}$ are the current magnitudes of the structure, diffusion, and physical geometry losses, respectively.

We then define the adaptive weights using the inverse proportional rule:

$$\alpha_i = \frac{1/\ell_i}{\sum_{j \in \{s,d,p\}} 1/\ell_j} \quad \text{for each } i \in \{s, d, p\} \tag{36}$$

This ensures that the sum of weights satisfies $\sum_i \alpha_i = 1$, while favoring tasks with smaller loss magnitudes by reducing their weight, and increasing the weight on larger-loss terms that need more attention.

The final total loss is as follows:

$$\mathcal{L}_{\text{total}}^{(t)} = \alpha_s \cdot \ell_s + \alpha_d \cdot \ell_d + \alpha_p \cdot \ell_p \tag{37}$$

where each term is adaptively scaled to counteract imbalance across sub-tasks.

## F.3 Empirical Results

We evaluate the impact of adaptive weighting in Table 5, comparing fixed scalar weights to our adaptive strategy. Notably, the adaptive scheme achieves the best performance across all evaluation metrics, demonstrating superior capability in fine-grained structure modeling.

Table 5: Experimental results on different weight adjustment strategies of the loss function. a: the weight of the structure term. b: the weight of the diffusion term. c: the weight of the physical and geometry term.

| Weight | | | RMSD ↓ | | Success Rate ↑ | |
|---|---|---|---|---|---|---|
| a | b | c | Overall | Side-chain | Drug <2Å | Overall <2Å |
| 1.0 | 1.0 | 1.0 | 2.22Å | 1.04Å | 49.13% | 51.32% |
| 0.5 | 1.0 | 1.5 | 2.43Å | 1.65Å | 45.27% | 49.48% |
| dynamic | dynamic | dynamic | **2.01Å** | **0.29Å** | **61.60%** | **65.30%** |

(↑) / (↓) denote higher / lower is better. Top 1 results are **bold**.

## G    Conformational ensemble docking performance

The heatmap in Figure 3 further emphasizes the consistency of DynaPhArM across structurally heterogeneous targets. The L-RMSD distributions across the DUDE27 benchmark illuminate core mechanistic divergences in how docking methods address receptor flexibility, conformational selection, and induced-fit coupling. DynaPhArM achieves the lowest L-RMSD in 18 out of 27 targets. By leveraging physics-constrained optimization, DynaPhArM resolves steric conflicts and captures long-range coupling, which is critical for targets with substantial loop or helix remodeling, such as BRAF (2.70 Å), where DFG-out to DFG-in transitions are necessary for accurate binding pose prediction.

In contrast, rigid-receptor docking methods like AutoDock and Glide systematically underperform on conformationally adaptive proteins. For example, in EGFR, the kinase C-helix undergoes a pronounced shift upon ligand engagement, a transition not accounted for by static scoring protocols, resulting in a marked deviation (AutoDock: 6.36 Å vs. DynaPhArM: 3.02 Å). These methods are intrinsically limited by their reliance on a single, often non-native, receptor conformation. TankBind demonstrates reasonable performance when structural homologs are available but degrade significantly on divergent folds or allosteric pockets (e.g., GRIA2: 10.33 Å), highlighting their dependency on prior alignment and conformational priors. While DiffDock demonstrates strong performance when docking to protein conformations that closely resemble those encountered during training, its ability to generalize deteriorates in extrapolative scenarios involving rare or unseen structural variations. This limitation is exemplified by the MET target, where DiffDock achieves a relatively high L-RMSD of 7.91 Å. The underlying issue stems from the model's reliance on implicit geometric representations learned from training data, which tend to favor commonly observed backbone and side-chain configurations. As a result, DiffDock often fails to accurately model rare but functionally important conformational states, such as the loop opening events observed in MET, due to insufficient exposure to such structural diversity during training. Hybrid approaches such as DiffBindFR, which integrate learned priors with restrained physics-based refinement, show moderate success in flexible regions (e.g., CDK2: 2.71 Å) but are still limited in sites requiring long-range conformational propagation, such as TGFR1 (3.61 Å), where allosteric activation involves distal loop dynamics and solvation effects. Fragment-based flexible methods like FlexPose capture local side-chain and loop adjustments (THRB: 3.49 Å), but are insufficient for large-scale domain or backbone reorganization.

## H    Case study on reconstructing accuracy

The study of Aceclofenac binding to the 6Y3C target structure carries important real-world implications for anti-inflammatory drug design. Aceclofenac is a widely used nonsteroidal anti-inflammatory drug [48], and 6Y3C corresponds to the crystal structure of cyclooxygenase-2 (COX-2), a crucial enzyme involved in prostaglandin synthesis and inflammation. Given the clinical relevance of COX-2 selectivity in minimizing gastrointestinal side effects while preserving therapeutic efficacy, accurate modeling of the binding interaction between Aceclofenac and 6Y3C is of paramount pharmaceutical interest. What makes this case particularly meaningful is due to the intricate nature of their binding mechanism, which presents significant challenges for structure reconstruction and pose prediction. Aceclofenac binds to COX-2 through a combination of hydrogen bonding, π−π stacking, and hy-

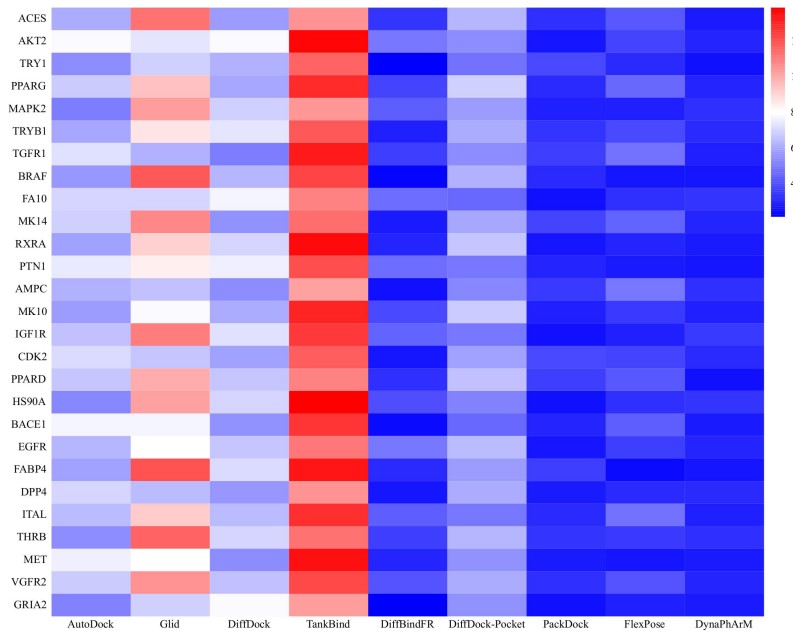

Figure 3: Performance of different methods in reconstructing complex structures within the DUDE27 test set, evaluated by L-RMSD.

drophobic interactions within a highly adaptive and conformationally flexible active site. The induced fit nature of COX-2, characterized by local rearrangements of the side chains (particularly in the hydrophobic channel and the T380, L384 and V423 residues), often leads to substantial deviations between the apo and holo states. These conformational changes, coupled with the drug's non-rigid scaffold and multiple rotatable bonds, introduce a high degree of uncertainty into conventional docking methods. By using this system, we aim to highlight our model's ability to capture such nuanced drug-induced conformational adaptations and accurately reconstruct the binding pose in a physically plausible manner, thus demonstrating its superior capacity to handle real-world flexible docking scenarios.

As shown in 4, DynaPhArM achieves the lowest L-RMSD and the best reconstruction performance comparing to the other methods. According to b in 4, the drug engages in a complex network of interactions with several key residues, including T380, M367, H422, V423, Q179, and L384, which collectively orchestrate the formation of a well-defined binding pocket, characterized not merely by static complementarity but by conformational plasticity, especially within the receptor's side-chain architectures. DynaPhArM demonstrates a profound understanding of this induced-fit paradigm by accurately capturing the side-chain reorganization necessary for stable drug accommodation. In particular, DynaPhArM predicts a significant conformational shift in V423, which subtly reorients to alleviate steric hindrance and permit favorable van der Waals and hydrophobic contacts with Aceclofenac's chlorophenyl and phenylacetic moieties.

In contrast, FlexPose, despite achieving a comparable RMSD, fails to resolve a critical steric clash between V423 and the drug, which highlights the limitations of RMSD as a sole metric, as FlexPose's predicted pose superficially resembles the native binding mode but lacks the microscopic interaction fidelity required for biological plausibility. The clash at V423 indicates FlexPose's insufficient modeling of side-chain entropy compensation.

Traditional metrics such as RMSD provide a coarse assessment of geometric alignment, they often obscure deeper deficiencies in interaction fidelity and physical plausibility. The above analysis reveals that accurately capturing side-chain flexibility and resolving steric conflicts—particularly at key residues like V423, which is essential for modeling biologically valid binding poses. DynaPhArM's superior performance stems not merely from pose prediction accuracy, but from its ability to model the energetic and structural consequences of induced fit at atomic resolution, which enables it to

discriminate between superficially correct and mechanistically correct solutions—a capability critical for drug design applications where small structural inaccuracies can lead to large functional errors.

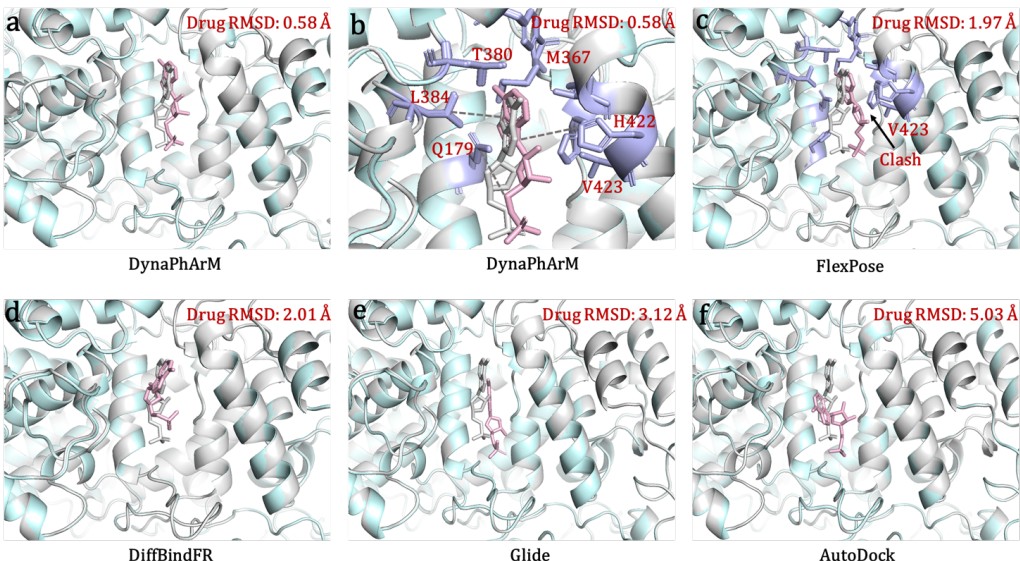

Figure 4: The case of different methods on the reconstructing the structure of the complex which is consist of the drug Aceclofenac and the target 6Y3C from PDB. **a, d-f** The comparative figures illustrating DynaPhArM, DiffBindFR, Glide and AutoDock predicted structure (cyon target cartoon, pink drug sticks) alongside the experimental structure in grey from PDB, respectively. **b** The detailed illustration of a, which shows the key residues (purple) in the binding pocket and the interaction forces between key residues and drugs in dashed lines. **c** The comparative figures illustrating FlexPose (the second best method) predicted structure (cyon target cartoon, pink drug sticks) alongside the experimental structure in grey from PDB, which also identified key residues in purple.

# I  Sensitivity Analysis of Priors

To evaluate the model's robustness to potential inaccuracies in priors, we conducted a sensitivity analysis on the pre-computed LJ matrix. To measure the degradation in model performance, we added Gaussian noise with an increasing standard deviation (Noise Level) to the LJ matrix, with results summarized in Table 6.

Table 6: Sensitivity analysis of the model to noise in the LJ prior. Performance is evaluated across different noise levels added to the LJ matrix.

| Noise Level | Overall RMSD (Å) ↓ | Side-chain RMSD (Å) ↓ | Ligand Success Rate (<2Å) ↑ |
|---|---|---|---|
| 0.0 | 2.01 | 0.29 | 61.60% |
| 1.0 | 2.07 | 0.38 | 58.10% |
| 2.0 | 2.10 | 0.49 | 54.20% |
| 5.0 | 2.21 | 0.55 | 53.17% |
| No Prior | 2.11 | 0.59 | 52.91% |

At low noise levels, performance changed very little, as the model's learned data-driven patterns and the adaptive $\gamma_i$ effectively compensated for minor inaccuracies.

At very high noise levels, the prior became actively misleading. The performance degraded, but it did not collapse entirely. The performance plateaued at a level comparable to that of the model trained without any prior, demonstrating that the model could learn to effectively ignore a completely useless prior.

In conclusion, the model is not overly sensitive to errors in the AutoDock-generated LJ potential and can effectively mitigate its impact.

# J   Details of Datasets

We have performed a detailed analysis of our full learnable dataset. The results summarized in Table 7, confirm that the dataset is sufficiently large and diverse to support the learning of generalized principles of protein-ligand interaction.

Table 7: Statistics of our curated dataset.

| Category | Metric | Value |
|---|---|---|
| Overall Dataset Size | Total Complexes | 21,762 |
| | Unique Protein Targets | 3,673 |
| Ligand Diversity | Mean Unique Ligands per Target | 6.57 |
| | Median Unique Ligands per Target | 3.0 |
| | Targets with > 5 Unique Ligands | 917 |
| | Targets with > 10 Unique Ligands | 444 |
| | Targets with > 20 Unique Ligands | 200 |
| Structural Diversity | Targets with > 5 Co-crystal Structures | 609 |
| | Targets with > 10 Co-crystal Structures | 318 |
| | Targets with > 20 Co-crystal Structures | 148 |
| Protein Family Diversity | HYDROLASE | 16.6% |
| | TRANSFERASE | 16.0% |
| | HYDROLASE/INHIBITOR | 10.3% |
| | TRANSFERASE/INHIBITOR | 10.1% |
| | TRANSCRIPTION | 3.6% |

In our evaluation of DynaPhArM's performance on the cross docking task, we compared it with a variety of baseline methods using several benchmark datasets, including CDK2, EGFR, DUDE27-HoloEns, Astex Non-Native (ANN), and CASF2016-Drug. The CDK2 and EGFR datasets are curated collections of high-resolution, inhibitor-bound protein–ligand complexes from the PDB [49]. The CDK2 set includes 11,317 Cyclin-Dependent Kinase 2 complexes, emphasizing ATP-competitive interactions in both rigid and flexible kinase active sites. EGFR comprises 67 wild-type and mutant kinase-inhibitor pairs, challenging algorithms to accommodate subtle shifts in the activation loop and $\alpha$C-helix. The DUDE27-HoloEns dataset [50] is a curated ensemble-docking benchmark comprising 27 DUD-E targets for which multiple holo structures have been experimentally determined. Each target's ensemble includes all available high-resolution co-crystal complexes, yielding 268 pairs of complexes. The ANN dataset [51] comprises 65 protein–ligand systems, each represented by a native holo complex and one or more non-native receptor conformations into which the cognate ligand is re-docked. Designed specifically for cross-docking, it challenges algorithms to recover the correct binding pose despite backbone and side-chain rearrangements not present in the ligand's original crystal structure, thereby gauging a method's robustness to receptor conformational variability. The CASF2016-Drug dataset is a refined subset of the widely used CASF2016 benchmark [52], focusing exclusively on drug-like ligands and their corresponding protein targets, and the original CASF2016 dataset comprises 65 distinct targets and 195 unique ligands. The curated subset emphasizes biologically and pharmacologically meaningful interactions, providing a more targeted benchmark for evaluating models in drug–target cross docking and structure-based virtual screening. The PoseBusters (PB) dataset [53] and toolkit form a comprehensive benchmark suite specifically designed to evaluate the structural plausibility and physical validity of protein–ligand complex models, which includes 428 high-quality crystal structures of protein–ligand complexes. By highlighting not only how close a predicted pose is, but also whether it makes chemical and physical sense. The detailed results on PB dataset can be found in Appendix L.

# K  Time consuming in reconstructing task

Our proposed method, DynaPhArM, achieves the fastest average reconstruction time (2.39 s per complex on an NVIDIA RTX 4090) by virtue of an integrated, end-to-end adaptive embedding framework that collapses what are traditionally multiple, disjointed pipeline stages into a single monolithic operator. Our approach fuses every stage from SMILES to 3D conversion and coarse docking to graph construction, encoding, quantization, multi-modal fusion, diffusion denoising, and coordinate regression into a single, zero-copy GPU pipeline. By embedding drug conformer generation and rough docking directly in GPU memory, we eliminate file I/O and CPU–GPU handoffs. A unified CUDA kernel then builds backbone, side-chain, and drug graphs in parallel, computing all scalar and vector features without repeated memory allocations or host transfers. We pack GNN inference, codebook distance computation, and embedding quantization into the same kernel launch on the fly—removing intermediate tensor copies. Cross-modal multi-head attention merges target and drug embeddings, which immediately flow into our diffusion noise/denoise network in one continuous stream. Finally, coordinate regression decodes the denoised embeddings in-place, avoiding reshape operations and extra kernel calls and eliminating redundant data transformations and minimizing CPU–GPU synchronization overhead.

In contrast, many existing approaches fragment the reconstruction task into discrete modules, each incurring its own latency. PackDock's exhaustive conformational sampling spawns a combinatorial explosion of rotamer states requiring successive scoring rounds, SurfDock's fine-grained surface tessellation enforces sequential mesh generation and probe-based mapping, and TankBind's dual-attention cross-correlation between ligand voxels and receptor grids introduces non-trivial dependency chains that stifle intra-kernel parallelism. DiffDock and DynamicBind, while streamlined for inference, nevertheless traverse multiple graph and volume-based passes, each demanding separate tensor allocations and synchronization barriers. FlexPose augments sampling agility with flexible-backbone energy minimization, but each iteration triggers a back propagation like recalculation of forces and gradients.

When DynaPhArM is deployed on the RTX 4090, it sustains throughput exceeding 1500 complexes per hour, opening new avenues for large-scale drug discovery campaigns. Its capacity to deliver sub-3-second reconstructions at near-experimental accuracy makes it particularly attractive for applications such as: (1) Ultra large virtual library screening, where millions of candidate compounds must be evaluated for plausible binding conformations; (2)Lead optimization loops, enabling rapid in silico iteration over substitution patterns and conformation-activity relationships; (3) Drug repurosing efforts, where existing pharmacopeia can be exhaustively reassessed against emerging targets with minimal computational investment; (4) Fragment-based drug discovery, facilitating real-time scoring of fragment assemblies during interactive fragment linking.

In sum, by harmonizing algorithmic depth with GPU-native efficiency, DynaPhArM promises to transform the computational throughput frontier, making true real-time, large-scale ligand screening not only feasible but routine.

# L  Structural rationality verification

## L.1  Evaluation metrics

Te PB success rate reflects the proportion of predicted complexes that meet key biophysical and geometric criteria evaluated by the PB toolkit, including steric clashes, bond geometry, hydrogen bonding, and conformational strain energy, thereby excluding physically implausible poses.

## L.2  Experiments results

As shwon in Figure 6, on PB dataset, traditional rigid-receptor dockers such as Glide, RosettaLigand, and AutoDock occupy the lower end of the spectrum, with overall success rates in the 25–35% range and PB success rates scarcely above 30%. TankBind's learned binding-site attention makes its performance achieve nearly 40% on PB dataset, yet it still suffers under the rigorous clash and contact checks imposed by PB toolkit. The DiffDock family marks a turning point: vanilla DiffDock climbs to 45% overall but only  38% PB success rate, whereas its pocket-aware variant breaches 50% overall and 45% PB success rate by conditioning on pre-identified cavities. DynamicBind and DiffBindFR further

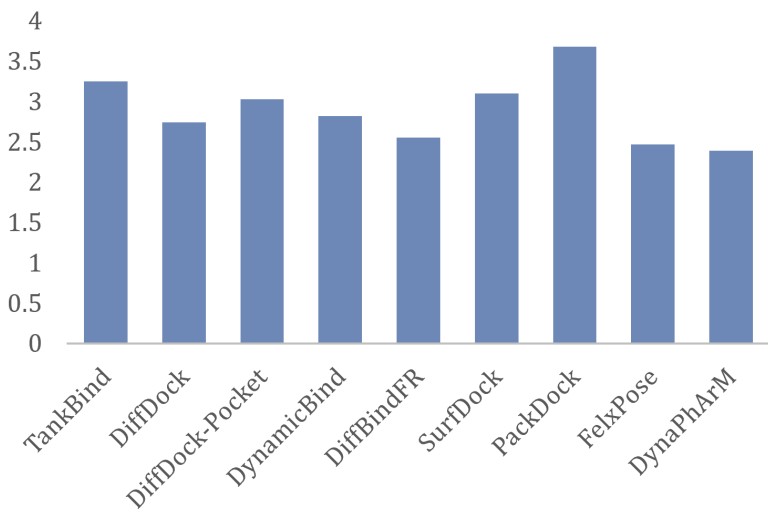

Figure 5: Speed comparison between DynaPhArM and other baselines.

refine pocket flexibility modeling and non-equilibrium binding pathways, respectively achieving 48–52% overall and 43–50% PB success rate. Surface-guided methods (SurfDock, PackDock) average 45% PB success rate by incorporating receptor surface complementarity, while FlexPose's hybrid sampling and filtering tactics push above 50% overall but still fall short under PB strict physical and geometric filters. DynaPhArM emerges at the 52% overall and 50% PB success rate), validating the power of combining generative diffusion with rigorous physics-informed penalties. The main reasons are in its diffusion-based generative core tightly coupled with differentiable geometry and physics constraints. The traditional rigid docking only jumps between a few discrete conformations, making it easy to fall into the trap of just getting stuck in the pocket or atomic collision. The diffusion process is a continuous and smooth random walk in a high-dimensional coordinate space, with each small step iteration allowing the system to explore feasible surrounding areas and correct minor collisions or geometric deviations layer by layer. During each reverse-diffusion denoising step, the model explores a continuous manifold of drug poses, guided not only by statistical priors learned from known complexes but also by softly enforced vdW, hydrogen-bond, and geometric consistency losses. DynaPhArM naturally avoids steric clashes and spurious contacts, steering the generative trajectory toward low-energy, physically plausible basins. As a result, it attains high reconstruction fidelity, precisely recapitulating native drug coordinates. This synergy between generative flexibility and physics-constrained regularization ensures DynaPhArM's poses consistently pass PB rigorous physical and geometric sanity checks.

## M  Limitations

Despite the promising results demonstrated by our proposed method, several limitations remain that warrant further investigation:

### M.1  Dependency on accurate structural inputs

Our model requires high-quality target protein structural inputs to effectively capture spatial and physical interactions. In real-world applications where target structures are predicted or noisy, the model's performance may degrade. Incorporating uncertainty-aware structural encodings or learning from sequence-only inputs could be potential extensions.

### M.2  Computational overhead of diffusion module

The diffusion denoising process introduces non-trivial computational complexity, especially during training. Although it enhances flexibility and robustness, this trade-off may limit scalability in high-throughput screening scenarios. Future work may explore more efficient generative backbones or lightweight denoising strategies.

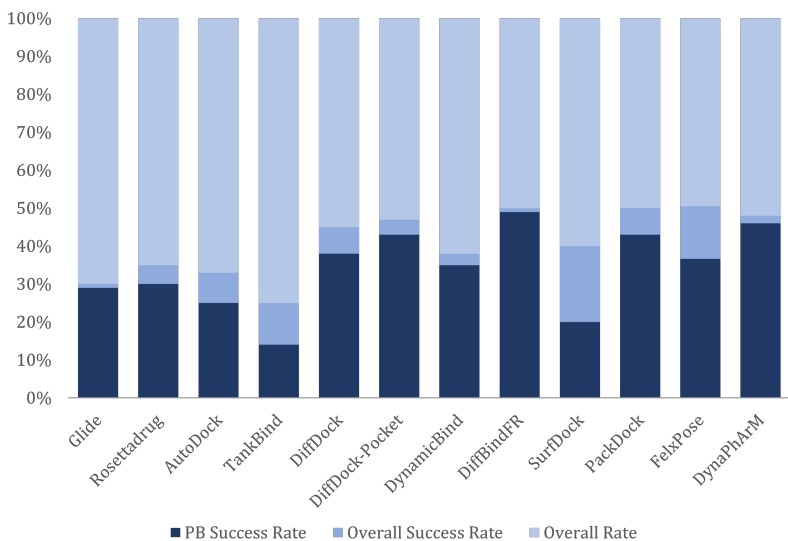

Figure 6: Performance of different methods in reconstructing complex structures within the Pose-Busters test set, evaluated by overall and PB success rate.

## M.3 Assumption of homogeneous task importance

While the adaptive weighting strategy dynamically balances learning signals, it assumes that all tasks are equally valuable in the normalized loss space. In practice, certain objectives (e.g., physics geometry-guided constraints) might be more critical for downstream performance. Incorporating task-priority priors or uncertainty-aware weighting schemes may offer finer control.

# N Experiment setting

The optimization of the model's parameters is performed using the Adam algorithm, as implemented in the PyTorch library. The optimizer is applied to the learnable parameters of several distinct architectural components, including the side-chain feature projector, the drug feature projector, the physics-based interaction module, the denoiser network within the diffusion module, and the SE(3)-equivariant graph transformer decoder. Adam computes individual adaptive learning rates for each parameter by utilizing estimates of the first and second moments of the gradients, which is particularly advantageous for a multifaceted model architecture such as the one employed, which integrates diverse modules requiring potentially different learning dynamics. The training process is initiated with a learning rate of $1 \times 10^{-4}$, while the $\beta_1$, $\beta_2$, and $\epsilon$ hyperparameters of the Adam optimizer are maintained at their default PyTorch values of 0.9, 0.999, and $1 \times 10^{-8}$, respectively.

Table 8: Key hyperparameters used in the proposed model, covering embedding dimensions, interaction modules, diffusion and decoder configuration, physical constraints, and inference setup.

| Name | Value | Description |
|---|---|---|
| D_E | 128 | Embedding dimension for backbone, side-chain, and drug features |
| D_K_INTERACTION | 128 | Key/Query/Value dimension |
| D_PRIME_INTERACTION | 256 | Output dimension of the physics-based interaction module, input to diffusion |
| MLP_HIDDEN_INTERACTION | 512 | Hidden dimension for MLP in the interaction module |
| N_ATTN_HEADS_INTERACTION | 4 | Number of attention heads |
| NUM_TRAIN_TIMESTEPS_DIFFUSION | 1000 | Total timesteps for the diffusion process during training |
| DENOISER_HIDDEN_DIFFUSION | 512 | Hidden dimension for the denoiser model |
| NUM_DECODER_LAYERS | 6 | Number of decoder layers |
| D_HIDDEN_ATTN_DECODER | 128 | Hidden dimension for attention in decoder |
| D_RBF_DECODER | 32 | Number of RBF kernels in decoder |
| D_MESSAGE_DECODER | 128 | Message dimension in decoder |
| D_MLP_HIDDEN_PHI_M_DECODER | 256 | Hidden dim. for MLP in message generation |
| D_MLP_HIDDEN_PSI_DECODER | 64 | Hidden dim. for MLP in coordinate update |
| LEARNING_RATE | 1e-4 | Learning rate |
| BATCH_SIZE | 64 | Batch size |
| NUM_EPOCHS | 50 | Number of epochs |
| REFINE_START_T_INFERENCE | 200 | Start timestep for denoising refinement |
| NUM_DENOISING_STEPS_INFERENCE | 50 | Steps in inference denoising loop |
| min_dist | 0.0 | Minimum distance for RBF |
| max_dist | 20.0 | Maximum distance for RBF |
| d_a_dist_max | 3.2 | Max donor–acceptor distance (H-bond) |
| d_a_dist_min | 2.5 | Min donor–acceptor distance (H-bond) |
| SIZE_X / SIZE_Y / SIZE_Z | 25 | Grid size in each spatial direction |
| NUM_MODES | 3 | Number of binding modes to generate |
| beta_start | 1e-4 | Start of noise schedule |
| beta_end | 0.02 | End of noise schedule |
| dropout_rate | 0.1 | Dropout rate |
| RMSD_THRESHOLD | 2.0 Å | Threshold for success (RMSD) |
| angle_min_deg | 120 | Minimum bond angle (H-bond) |
| angle_max_deg | 180 | Maximum bond angle (H-bond) |
| multiprocessing.num_workers | 64 | Number of worker processes for parallel tasks |
| multiprocessing.chunk_size | 10 | Chunk size for multiprocessing pool operations |
| drug_preparation.charge_model | gasteiger | Charge model used for ligand preparation |
| dataset_split.cd_hit_identity | 0.9 | Sequence identity threshold for CD-HIT clustering |
| dataset_split.validation_set_ratio | 0.2 | Proportion of data for the validation set |
| dataset_split.random_seed | 42 | Random seed for data splitting |
| backbone_encoder_params.num_layers | 4 | Number of layers in the backbone encoder |
| backbone_encoder_params.l_max_sh | 2 | Max degree for spherical harmonics |

