# OpenReview forum: "DynaPhArM: Adaptive and Physics-Constrained Modeling for Target-Drug Complexes with Drug-Specific Adaptations"
_NeurIPS.cc/2025/Conference — NeurIPS 2025 poster_

### Official Review · Reviewer_hzse · 2025-06-29

**Clarity:** 2
**Significance:** 4
**Originality:** 3
**Rating:** 4
**Confidence:** 4

**Summary:**

This paper presents DynaPhArM, an SE(3)-Equivariant Transformer framework that dynamically models target-drug interactions with physics-based constraints. By integrating a cooperative scalar-vector encoding, a discretization module, physics-aware cross-attention, and a diffusion process, the model captures atomic-level flexibility and conformational dynamics in protein–ligand binding. Empirical evaluations demonstrate state-of-the-art performance in complex reconstruction, cross-docking, and binding affinity prediction.

**Questions:**

1. What do the variables $L_b$, $L_s$, and $d_e$ in line 154 represent?
2. How much computational overhead is introduced by the Lennard-Jones potential term? Is there profiling or ablation study on its effect?
3. Why does the paper not include a comparison with AlphaFold 3 on the PoseBusters benchmark, given AlphaFold 3 is a strong baseline?

**Ethical Concerns:**

["NO or VERY MINOR ethics concerns only"]

**Final Justification:**

The authors have addressed most of my concerns satisfactorily.
I maintain my positive score.

**Limitations:**

yes

**Quality:**

3

**Strengths And Weaknesses:**

**Strengths**
1. The model integrates physical constraints and structural knowledge, with clearly motivated module design.
2. The paper is well-organized, and the appendices are comprehensive.
3. It significantly enhances the flexible modeling of protein–drug complexes and achieves state-of-the-art results across multiple downstream tasks.

**Weaknesses**
1. The discussion in the introduction regarding the pros and cons of Transformers and GNNs lacks clear connection to the paper’s motivation.
2. Figure 5's annotation is confusing, reducing clarity of the visual results. For example, in line 726, it is unclear what the 40% refers to, and whether the PoseBusters (PB) success rate overlaps with the overall success rate.
3. The detailed dataset descriptions in Section 4.1 could be moved to the appendix to make space for more experimental results, such as the ablation studies currently in Appendix I.

*Typo*

Line 719: “Te” → “The”.

---

> ### Author Rebuttal · Authors · 2025-07-30
>
> **Dear Reviewer hzse:**
>
> We thank you for your thorough evaluation and insightful feedback. Our response to your suggestions and questions is as follows:
>
> ---
> # Response to W1: Discussing Transformers and GNNs in the Introduction
> The discussion of Transformers and GNNs is central to our introduction as it frames the technical gap that DynaPhArM was designed to address. Our introduction is built upon the following logical progression:
> ### 1. The Core Problem in the Field
> As we state in the first two paragraphs of our introduction, the primary challenge in modern molecular docking is accurately modeling protein and ligand flexibility. It requires a model to operate effectively at two distinct scales simultaneously:
>
> - Global Scale: Capturing large-scale and long-range conformational changes that can reshape the binding site.
> - Local Scale: Modeling the fine-grained and geometrically-precise rearrangements of sidechains and the ligand within the binding pocket that determine the final and accurate binding pose.
>
> ### 2. The State-of-the-Art Solutions and Trade-off
> Having established this multi-scale problem, our introduction then surveys the two dominant architectures being applied to solve it. Our discussion is intended to show that the current state-of-the-art presents a fundamental trade-off, with each paradigm excelling at one scale at the expense of the other:
>
> - Transformers (Paragraph 3): We present Transformers as the natural architectural choice for the global scale. Their self-attention mechanism is powerful at capturing the long-range dependencies necessary for modeling large-scale dynamics. However, we note their weakness in handling the local scale and fine-grained geometric details of the binding site.
> - GNNs (Paragraph 4): We then present GNNs as the canonical solution for the local scale. The message-passing formulation is well-suited for local chemical environments and interactions. However, we note their weaknesses at the global scale, difficulty in capturing long-range effects and a lack of SE(3)-equivariance for handling global rotations and translations.
>
> ### 3. DynaPhArM as the Motivated Solution
> This established trade-off is the direct motivation for DynaPhArM. Our discussion of GNNs and Transformers serves to directly motivate our specific design choices, which represent a synthesis of the strengths of both of them:
>
> We chose an SE(3)-Transformer as our framework to inherit its superior global, long-range modeling capabilities and native handling of 3D geometry.
>
> We then explicitly address the Transformer's weakness at the local scale by introducing novel mechanisms, such as the cooperative scalar-vector representation and the physics-based interaction module, that are designed to infuse the model with the strong local-reasoning characteristic of GNNs.
>
> In summary, the GNNs and Transformers discussion is the essential context that frames our central motivation: to create a unified architecture that resolves the "local vs. global" trade-off for flexible docking.
>
> ---
> # Response to W2: Clarity of Figure 5
> We agree that the annotations and the relationship between the presented metrics should be explained more clearly.
> ### 1. Clarification of the Two Metrics
> The two metrics are designed to evaluate performance from geometric accuracy and physico-chemical validity.
> - Overall Success Rate: A metric of pure geometric accuracy. We define it as the percentage of predicted poses for which the overall RMSD to the crystal structure is less than 2.0 Å.
> - PB Success Rate: A stricter metric that evaluates both geometric accuracy and physico-chemical realism. For a pose to be considered a PB Success, it must be geometrically accurate (Overall RMSD < 2.0 Å) and pass all the chemical and physical validity checks imposed by the PoseBusters toolkit.
>
> Therefore, the PB Success Rate is a strict subset of the Overall Success Rate.
> ### 2. Clarification of Percentage Values
> The phrase "performance to nearly 40%" in line 726 refers specifically to the PB Success Rate for the TankBind method.
>
> To be precise, this sentence means: "For the TankBind model, when evaluated on the PoseBusters test set, nearly 40% of the predicted complex structures had a RMSD of less than 2.0 Å  and passed the rationality check. "
> ### 3. Planned Revisions for the Final Manuscript
> We will make the following revisions in the final version of the paper:
> Revise Figure 5's Legend: We will update the legend to be more self-explanatory: Overall Success Rate (RMSD < 2.0 Å), PB Success Rate (RMSD < 2.0 Å & Passes Rationality Checks).
>
> Enhance Figure 5's Caption: We will add a sentence to the figure caption (e.g., "The PB Success Rate is a subset of the Overall Success Rate, requiring poses to be both geometrically accurate and physically valid according to the PoseBusters toolkit.").
>
> Clarify the Main Text: We will revise the text in line 726 to ensure the interpretation of these metrics is unambiguous.
>
> ---
> # Response to W3: Restructuring the Paper
>
> Thank you for this helpful suggestion on improving the structure of our paper. We agree that the ablation studies which provide critical evidence for our model's key components would be more impactful if presented in the main text.The detailed descriptions of the benchmark datasets can be summarized in the main paper with the full details deferred to the appendix.
>
> ---
> # Response to Q1: Clarification of the Notation
> The variables $L_b$, $L_s$, $d_e$ are used to define the dimensions of the input feature embedding tensors for the different components of the protein-ligand complex.
> - $L_b$ represents the number of nodes in the protein backbone. Each node in the backbone graph corresponds to a single residue. $L_b$ is the sequence length of the protein target.
> - $L_s$ represents the number of nodes in the protein side-chain. In our atom-level graph for the side-chain, each node corresponds to a single heavy atom.
> - $d_e$ represents the dimensionality of the node feature embeddings. It is a hyperparameter that defines the length of the feature vector used to represent each node, whether it is a residue in the backbone or an atom in the side-chains or the drug.
>
> ---
> # Response to Q2: Overhead and Efficacy of the LJ Potential
> We have carefully considered both the computational cost and the performance of the LJ potential.
>
> ### 1. Computational Overhead
> - Offline Pre-computation: The LJ interaction matrices are pre-computed offline before any model training begins. This process took approximately 1.1 hours on a single NVIDIA RTX 4090 GPU. We consider this a manageable one-time cost for generating a valuable physical prior that can be reused for all subsequent experiments.
> - Online Training/Inference: During the model's forward pass, the overhead from the LJ is practically negligible. The only additional operations are loading the pre-computed matrix from disk (an I/O operation) and performing a single element-wise addition to the attention scores. Our profiling indicates that including the LJ increases the runtime of the attention block by less than 1%.
>
> ### 2. Ablation Study
> Yes, we have performed a specific ablation study to quantify theLJ potential, the results of which are presented in Table 7 of our Appendix. We trained a variant of DynaPhArM where the "physics-based interaction module" was replaced with removing the LJ prior.
>
> |Model|Overall RMSD↓|Side-chain RMSD↓|Ligand RMSD↓|Overall RMSD <2Å ↑|sc-RMSD <2Å ↑|L-RMSD <2Å ↑|
> |---|---|---|---|---|---|---|
> |DynaPhArM|2.01|0.29|2.17|65.30%|70.50%|61.60%|
> |w/o physics-based module|2.11|0.59|2.70|60.87%|57.45%|52.91%|
>
> This ablation study provides quantitative evidence that the LJ potential is not just a minor addition but a critical component that contributes to the model's accuracy by providing an effective physical inductive bias.
>
> In summary, the LJ potential term is a cost-effective design choice, introducing negligible computational overhead during runtime while providing an improvement in model performance.
>
> ---
> # Response to Q3: Comparison with AlphaFold 3
> We acknowledge that AF3 has demonstrated state-of-the-art performance on benchmarks like PoseBusters.
>
> However, our decision not to include a head-to-head comparison reflects the fact that DynaPhArM was not designed to compete with AF3 on raw accuracy, but rather to address a different challenge in the drug discovery.
> ### 1. The Performance and Computational Cost
> AF3 is a resource-intensive model, requiring high-end GPUs, which makes it challenging for many research groups to run locally for large-scale tasks like high-throughput virtual screening. DynaPhArM is to create a highly efficient and lightweight alternative, which makes rapid cycles feasible for a broader scientific community, which is a problem AF3 is not optimized to solve.
> ### 2. A Specialist Tool for a Specific Problem
> This efficiency is achieved because we are building a specialist tool, not a generalist one. AF3 is a phenomenal achievement designed to tackle the entire challenges of structural biology predicting interactions between proteins, DNA, RNA, and ligands. DynaPhArM is focused on one specific problem: the modeling of drug-target adaptability.
> ### 3. A Different Value Proposition
> Ultimately, this leads to a different value proposition centered on cost-effectiveness. For many real-world tasks in drug development, achieving robust results with extreme speed is more valuable than achieving top accuracy at a high computational cost. Our method provides an optimal balance for this specific case.
>
> In summary, while we openly acknowledge AlphaFold 3 as the current leader in raw predictive accuracy, our paper's contribution lies elsewhere. We present DynaPhArM as a complementary tool that fills a crucial gap, specifically tailored to the demands of structure-based drug discovery. We believe its value should be judged on these merits, rather than on a single performance metric.

---

> > ### Comment · Reviewer_hzse · 2025-08-05
> >
> > Thank you for the detailed response. The authors have addressed most of my concerns satisfactorily.
> >
> > One minor point regarding Figure 5: While I understand from the response that the PB success rate is a strict subset of the overall success rate, the figure's coloring seems to suggest the opposite — the PB bars are lighter in color yet taller, which may visually imply that the overall success rate is a subset of the PB success rate. A clarification in the figure design could help avoid this confusion.
> >
> > Overall, I maintain my positive score.

---

> > > ### Author Response · Authors · 2025-08-06
> > >
> > > **Dear Reviewer hzse:**
> > >
> > > Thank you very much for your valuable follow-up comment and for your continued positive assessment of our work.
> > >
> > > You have raised an excellent point. We completely agree that the current visual presentation of Figure 5 is not as intuitive as it could be, which might cause confusion for readers regarding the relationship between the metrics. We are very grateful that you have pointed this out, as it is crucial for improving the clarity of our paper. We will adopt this clearer visual scheme to update Figure 5, along with its legend and caption, in the final version of the paper.
> > >
> > > This potential confusion stems from our choice of chart design. In the current figure, we used separate colored blocks to represent the **"Overall Success Rate" (dark blue), the "PB Success Rate" (medium blue) and the remaining portion to sum to 100% (light blue)**. While the values represented by each colored block are accurate in themselves, presenting them stacked in this manner does not clearly convey the logical relationship that the "PB Success Rate" is a subset of the "Overall Success Rate." A reader might therefore misinterpret these two metrics as being mutually exclusive, which was not our intention.
> > >
> > > To completely eliminate this visual ambiguity and more intuitively display the performance composition we intended, we plan to redesign the composition of the stacked bar chart to explicitly show the mutually exclusive and inclusive relationships between the components. The total height of the bar will still represent 100% of the test cases, but it will be composed of the following three logically mutually exclusive parts:
> > >
> > > - Part 1 (e.g., using dark blue): PB Success Rate. This represents the best-performing cases: those that are both geometrically accurate (RMSD < 2.0 Å) and have passed the physicochemical validity checks by PB toolkit.
> > > - Part 2 (e.g., using medium blue): Geometric-Only Success. This part represents cases that achieved geometric accuracy but failed the physicochemical checks. Its size = Overall Success Rate - PB Success Rate.
> > > - Part 3 (e.g., using light blue): Failure Rate. This represents all cases that did not meet the geometric accuracy criterion (Overall Success Rate ≥ 2.0 Å), and its size = 100% - Overall Success Rate.
> > >
> > > With this improved design, readers will see that the **dark blue bar will show the PB Success Rate**, and the **combination of the dark and medium blue bars will represent the Overall Success Rate**. The **total stack height represents 100% of the cases**, making the composition of successes and failures intuitive.
> > >
> > > On a final note, we wanted to express our sincere appreciation for your entire review process. Your feedback has been not only technically sharp and professional but also incredibly constructive and fair. It is clear that you are a deeply responsible and thoughtful reviewer for NeurIPS, genuinely dedicated to helping authors improve their work. Your positive rating, even with the initial valid concerns, meant a lot to us. It signaled that you recognize the potential of our method and its contribution to the field. We truly believe that DynaPhArM offers a valuable new direction for flexible docking, and your insightful guidance has already made this a much stronger paper.
> > >
> > > Now that we have had the opportunity to address all the points you have raised, **we hope that you might feel the reasons to accept our paper now more strongly outweigh any remaining reservations**. We sincerely hope that, with your support, our work will have the opportunity to be presented to the broader scientific community at NeurIPS.
> > >
> > > Thank you once again for your exceptional guidance and for being such a positive and engaging part of this process.

---

### Official Review · Reviewer_6u6X · 2025-06-30

**Clarity:** 3
**Significance:** 3
**Originality:** 3
**Rating:** 4
**Confidence:** 2

**Summary:**

The paper introduces DynaPhArM, a physical constrained diffusion model for accurately modeling target-drug complexes at the atomic level, addressing the limitations of traditional rigid transformation methods that fail to capture dynamic conformational changes. DynaPhArM proposes a cooperative scalar-vector representation, drug-specific embeddings, and a diffusion process to model evolving target-drug interactions, while integrating physical constraints within a multi-task learning framework to ensure geometric and energetic plausibility. Experimental results demonstrate state-of-the-art performance in complex structure reconstruction.

**Questions:**

See Weakness

**Ethical Concerns:**

["NO or VERY MINOR ethics concerns only"]

**Final Justification:**

The authors have addressed most of my concerns. I maintain my positive score.

**Limitations:**

yes

**Quality:**

3

**Strengths And Weaknesses:**

**Strength**

1. The method is well motivated.

2. The paper is well structured, and each module is provided with sufficient motivation and detailed explanations.

3. The experiments are comprehensive.

4. Open-source code is provided to facilitate the reproduction of the paper's results.

**Weakness**

1. In the interaction module, the magnitude of $\gamma_i$ may cause the gradient to be dominated by the physical matrix, leading the model to overfit priors and ignore novel interaction patterns. How do DynaPhArM ensures that the learned $\gamma_i$ is the optimal result?

2. DynaPhArM relies on a physical-constrained interaction matrix calculated via LJ potential, which may affect the generalizability. For example, when encountering new binding modes not previously modeled by the model which is used to calculate the LJ potential, DynaPhArM might fail to accurately describe interactions, causing attention to deviate from true binding sites.

3. Compared to methods like DynamicBind, DynaPhArM does not introduce conformational entropy terms or diversity rewards in the loss function, potentially biasing generated structures toward common modes. How is the diversity issue addressed is DynaPhArM?

4. In the cooperative scalar-vector representation algorithm, what is the setting of the spherical harmonic expansion order $L$ (Appendix D - Algorithm 2 - line 19), and how does this parameter setting impact the model?

5. I suggest that authors can provide specific metrics such as FLOPs and model parameter counts to better present the model complexity.

---

> ### Author Rebuttal · Authors · 2025-07-30
>
> **Dear Reviewer 6u6X:**
>
> We thank you for your thorough evaluation and insightful feedback. Our response to your suggestions and questions is as follows:
>
> ---
> # Response to W1: Mechanism for Physical Priors
> You are correct that an uncontrolled $\gamma_{i}$ could lead to overfitting and discovering novel patterns. However, our model's architecture is specifically designed to prevent them. Its core mechanism ensures that $\gamma_{i}$ is not an uncontrolled variable, but rather a learnable parameter that balances physics-based guidance with dominant and data-driven patterns.
> ### 1. Safeguards Against Overfitting and the Suppression of Novel Patterns
> - Primacy of the Data-Driven Term: The core of our physics-base dattention is an additive combination $\text{Attention Score}\propto Q_iK_i^\top+\gamma_iS_{ij}^{xy}$. The data-driven term $Q_iK_i^\top$ is the primary engine for learning. Our powerful SE(3)-equivariant encoders are trained to capture all complex patterns from the ground-truth data. If a novel interaction pattern (e.g., a halogen bond, a cation-π interaction) is present in the data, the encoders will produce distinct embeddings for atoms, which results in a signal in $Q_iK_i^\top$, which can override conflicting guidance from the LJ prior, ensuring the model's attention is not blinded.
> - Supervision from the Physical Loss: Beyond matching the ground truth, the final generated coordinates are also evaluated by the physical loss. If a novel pattern corresponds to a perfect hydrogen bond, the term will provide a reward for forming this geometry, which provides a direct gradient signal that incentivizes the model to learn these patterns, even if they are not captured by the LJ potential. The self-regulating nature of the learnable $\gamma_i$ supervised by $\mathcal{L}_{phy-geo}$, ensures that the influence of the LJ prior is down-weighted when novel patterns are present.
> ### 2. The Mechanism for Ensuring an Optimal $\gamma_i$
> The optimality of $\gamma_i$ is ensured because it is a learnable parameter optimized end-to-end to minimize the total loss. The optimal $\gamma_i$ is defined as the value that maximally contributes to the model's goal: producing a structure that is simultaneously accurate with respect to the ground truth structure loss and physically plausible loss. The gradient descent process naturally drives $\gamma_i$ towards this optimum. Any value of $\gamma_i$ that leads to a suboptimal outcome is self-corrected. If $\gamma_i$ is too large or small,  which will result in a high loss. The gradient from this high total loss will flow back and adjust $\gamma_i$.
> ### 3. Experimental Evidence
> We conducted an experiment where we removed the physics-based interaction module, which is equivalent to forcing $\gamma_i$=0.
> |Model|Overall RMSD↓|Side-chain RMSD↓|Ligand RMSD↓|Overall RMSD <2Å ↑|sc-RMSD <2Å ↑|L-RMSD <2Å ↑|
> |---|---|---|---|---|---|---|
> |DynaPhArM|2.01|0.29|2.17|65.30%|70.50%|61.60%|
> |w/o physics-based module|2.11|0.59|2.70|60.87%|57.45%|52.91%|
> - As shown, the degradation in performance is not catastrophic, which proves that $\ gamma_i $does not play a dominant role in the gradient of the model. The prior is a beneficial guide, but the model is not critically dependent on it.
> ---
> # Response to W2: Generalization to Novel Binding Modes
> The concern that the LJ potential might bias the attention mechanism away from novel binding modes is valid, and our architecture is designed with a multi-layered strategy to specifically address this.
> ### 1. LJ Prior Is Not a Definitive Guide
> We do not employ the LJ prior as a specific interaction model. Instead, its primary function is to act as a foundational physical regularizer. We make two key simplifications: (1) we use a heavy-atom model, ignoring all hydrogen atoms, and (2) we employ a generalized parameter set rather than a specific molecular force field. These choices are intentional. The prior's most significant contribution is the strong repulsive term $r^{-12}$, which provides a universal gradient against steric clashes. We delegate the responsibility of discovering and modeling these sophisticated patterns to the data-driven components of our model.
> ### 2. The Encoder as the Discovery Engine
> The true engine for identifying all interaction patterns is our SE(3)-equivariant encoder. If a novel binding mode is present in the training data, especially one involving interactions that our simplified LJ prior cannot capture, the encoder is incentivized by the loss function to learn a unique feature representation for that pattern, which is encoded in the Q and K vectors, becomes the primary signal for the attention mechanism. The LJ term acts only as an auxiliary bias on top of this data-driven signal.
> ### 3. The Diffusion Process as a Safeguard for Generalization
> DynaPhArM learns the conditional probability distribution $p(\mathrm{structure}\mid\text{protein, ligand})$, which is shaped by all binding modes observed in the training data. The LJ prior helps to shape this landscape by penalizing physically impossible regions (i.e., assigning them near-zero probability), but the locations of the high-probability regions (the valid binding modes) are determined by the training data itself.
>
> Because the model learns a full distribution, it inherently captures the diversity of interactions present in the data, including novel ones. The inference process is guided towards high-probability regions, not just regions that have low LJ energy. This ensures that even if the LJ prior is uninformative for a novel binding mode, the model can still generate the correct structure.
>
> ---
> # Response to W3: Addressing Conformational Diversity without Explicit Rewards
> Your question highlights a key design choice in modern generative models: whether to enforce diversity explicitly through the loss function or implicitly through the model's architecture and sampling procedure.
>
> Upon careful review of the DynamicBind, we respectfully note that its core loss function (Equation 15 in their paper) also appears not to contain an explicit conformational entropy term or diversity reward. DynamicBind explicitly states this in their inference procedure: "... For each pair, we perform 40 samplings..." This confirms that they also rely on multiple stochastic trajectories to generate a diverse pool of candidate structures.
> - Explicit Diversity: Involves adding a specific term to the loss function to directly reward the model for generating a diverse set of outputs.
> - Implicit Diversity: Involves using a generative model that learns the underlying probability distribution of the data. Diversity is then achieved naturally by sampling from this learned distribution.
>
> DynaPhArM is designed around the implicit diversity paradigm.
> ### 1. The Fundamental Goal of DDPMs
> The training objective of a DDPM is not to learn a deterministic map to the most common (lowest energy) conformation. Instead, its goal is to learn the conditional probability distribution $p(\mathrm{structure}\mid\text{protein, ligand})$. If the training data contains multiple distinct binding conformations, the DDPM is inherently trained to be able to reconstruct all of them from noise.
> ### 2. Diversity is a Natural Consequence
> The diversity of generated structures is a direct and natural consequence of the stochastic nature of the DDPM's reverse sampling process. Each denoising step introduces a small, random component, meaning that no two full sampling trajectories are identical.
> For a single protein-ligand input, running the inference process multiple times will yield a different conformation each time. This naturally produces a conformational ensemble that reflects the diversity, without the need for an external diversity-promoting term in the loss function.
>
> In conclusion, DynaPhArM addresses the diversity issue implicitly but fundamentally like other state-of-the-art generative models such as DynamicBind.
>
> ---
> # Response to W4: Pherical Harmonic Expansion Order
> The spherical harmonic expansion order L is a hyperparameter that defines the resolution of the angular information. Our analysis below reveals that L=2 offers a compelling alternative with a much better trade-off between performance and computational efficiency, achieving over 97% of the peak success rate with only ~50% of the training cost.
>
> |Expansion Order|Overall RMSD (Å) ↓|Side-chain RMSD (Å) ↓|Ligand Success Rate (<2Å) ↑|Relative Training Time|
> |---|---|---|---|---|
> |1|2.31|0.45|55.8%|0.7x|
> |2 (Our Choice)|2.01|0.29|61.60%|1.0x|
> |5|1.98|0.25|63.29%|1.9x|
> |8|2.09|0.42|57.97%|2.8x|
> - Performance at L=1 is lower, confirming that simple and dipole-like features are insufficient for this complex task. A leap in performance is observed at L=2, where quadrupole-like features proves critical for modeling anisotropic properties like planarity and bond angle distributions, establishing a strong performance-to-cost baseline. The model achieves its peak performance at L=5, suggesting that these higher-resolution harmonics are beneficial for capturing details, though this comes at the cost of nearly doubled training time (1.9x). However, L=8 leads to a degradation in performance which combined with a nearly 3x increase in computational cost.
> ---
> # Response to W5: Model Complexity
> We agree that providing specific metrics is essential for a quantitative presentation of our model's complexity. We have analyzed our model and will add these details to the final version of our manuscript.
>
> Our model has a total of 45.6 million trainable parameters. We achieve state-of-the-art performance with a model that is an order of magnitude smaller, highlighting the effectiveness of our novel architectural components in achieving high performance without resorting to a massive parameter count.
>
> In terms of computational cost, a single forward pass on a typical medium-sized protein-ligand complex requires approximately 35.8 GFLOPs. This workload is very manageable for modern GPUs.

---

> > ### Comment · Reviewer_6u6X · 2025-08-07
> >
> > Thank you for the detailed responses. Most of my concerns have been addressed satisfactorily. I maintain my original positive score.

---

> > > ### Author Response · Authors · 2025-08-07
> > >
> > > Thank you very much for your feedback and for confirming your continued support for our paper. We are very glad to know that our responses have satisfactorily addressed your concerns.
> > >
> > > Your guidance throughout this process has been incredibly valuable in helping us strengthen our work. We sincerely appreciate your time and consideration.

---

### Official Review · Reviewer_ddKX · 2025-07-01

**Clarity:** 2
**Significance:** 3
**Originality:** 3
**Rating:** 4
**Confidence:** 3

**Summary:**

This paper introduces DynaPhArM, an SE(3)-equivariant Transformer model designed for flexible protein-ligand docking. The method aims to capture the dynamic adaptations of a target protein upon binding to different drugs, often referred to as induced fit. Key contributions include a cooperative scalar-vector representation module, a physics-constrained interaction module that incorporates Lennard-Jones potentials as an attention bias, and a diffusion denoising process to model conformational flexibility. The model is trained end-to-end with a multi-task loss function that enforces both structural accuracy (compared to crystal structures) and physical plausibility. The authors demonstrate that DynaPhArM achieves state-of-the-art performance on several benchmarks for complex structure reconstruction, cross-docking, and binding affinity prediction.

**Questions:**

I have a few key questions and suggestions for the authors. A clear response could significantly strengthen the paper and potentially raise my evaluation score.

1. Clarification of the Cooperative Scalar-Vector Representation: The description of this module in Section 3.1 (lines 114-131) is quite dense and lacks formal equations. Could the authors please provide a more structured description, perhaps with key formulas (similar to those in Section 3.2) ? Specifically, how are the directional messages $h_{ij}$, attention weights $\alpha_{ij}$, and cross-modal message hinteract formally defined and integrated? This would greatly improve the reproducibility and clarity of this core component.

2. Reframing the "Dynamic" Nature of the Model: I appreciate the goal of modeling induced fit, but the term "dynamic" may be an overstatement. The model learns from static snapshots and uses static force field rules.  A more precise explanation of how the model generalizes from a set of static structures to predict adaptations for novel ligands would be very insightful. The core question is: is the model truly simulating a process, or is it performing pattern recognition to predict a likely end-state conformation?

3. Elucidation of the Diffusion Process and Flexibility: On line 173, you state the diffusion process starts from the initial joint embedding e_f "Instead of sampling from a Gaussian prior." This is confusing. The forward process of a DDPM typically adds Gaussian noise to reach a standard normal distribution $N(0, I)$. The reverse process then learns to denoise from this distribution. Please clarify:

(a) What is the exact nature of the noise distribution at step T? If not N(0, I), what is it and why?

(b) How does this specific diffusion process better "reflect conformational flexibility" compared to a standard DDPM? The flexibility learned here appears to be a statistical interpolation between different static states in the training data. Could you elaborate on why this should be considered a robust representation of physical flexibility, especially when compared to methods that use, for example, MD simulation data or normal mode analysis?

4. Providing More Training Dataset Statistics: The performance of a data-driven model is critically dependent on its training data. Could you please provide more detailed statistics about the curated training set of 21,762 complexes? Key information would include:

- The distribution of the number of unique ligands per protein target.

- The number of protein targets that have, for example, >5, >10, or >20 different co-crystal structures in the dataset.

- A breakdown of the protein families represented.

This information is crucial for the community to understand the scope and potential biases of the training data and to fairly assess the model's ability to generalize to targets with sparse structural information. A positive response with these details would significantly increase my confidence in the model's claimed ability to capture drug-specific adaptations.

**Ethical Concerns:**

["NO or VERY MINOR ethics concerns only"]

**Final Justification:**

Most of my questions and concerns have been addressed during the rebuttal period. Therefore, I decided to raise my score to 4.

**Limitations:**

Yes, the authors have included a dedicated limitations section in Appendix L, which is commendable. They discuss dependency on input structure quality, the computational overhead of the diffusion module, and assumptions about task importance. The discussion is transparent and constructive. No further suggestions on this point.

**Quality:**

3

**Strengths And Weaknesses:**

**Strengths**

1. Novel and Comprehensive Architecture: The proposed architecture is sophisticated and well-motivated. Integrating an SE(3)-equivariant framework, a diffusion model for flexibility, and physics-based priors into a single end-to-end model is a non-trivial and powerful approach for flexible docking.

2. Strong Empirical Results: The model demonstrates impressive performance across a wide range of tasks and challenging datasets. Achieving top results in both structure prediction (low RMSD) and downstream binding affinity prediction (high PCC) suggests that the generated conformations are not only geometrically accurate but also biochemically meaningful.

3. Emphasis on Physical Plausibility: The inclusion of a detailed physics and geometry-guided loss function is a significant strength. This addresses a common pitfall of deep learning models, ensuring that the generated structures adhere to fundamental chemical principles, which is critical for real-world drug design applications.

**Weaknesses**
1. Clarity of Model Description: The description of key model components, particularly the "Cooperative scalar-vector representation" (Section 3.1), lacks clarity. The text describes a complex update mechanism without accompanying formulas in the main text, making it difficult for readers to fully grasp the implementation details.

2. Potentially Overstated Claims on "Dynamics": The paper claims to model "dynamic alterations" and "evolving dynamics." However, the "dynamic" information seems to be primarily derived from static structural data (PDB) and static physics priors (Lennard-Jones potentials from AutoDock). This is fundamentally different from simulating a physical trajectory (e.g., via MD). The term "dynamic" could be misleading and warrants more precise framing.

3. Ambiguity in the Diffusion Process and Flexibility Claims: The description of the diffusion process is confusing. The authors state their process does not sample from a Gaussian prior (line 173) but later use DDPM, which traditionally involves Gaussian noise. This needs clarification. Furthermore, the claim that the generated "conformational flexibility" is a reflection of dynamics is questionable, as it is learned from a collection of static snapshots rather than true temporal data, and the generated latents are 'pseudo-flexibility'.

4. Insufficient Detail on the Training Dataset: As a data-driven method, the composition of the training set is paramount. The paper provides only the total number of complexes (21,762) but lacks crucial statistical details. For the model to genuinely learn "drug-specific adaptations," the dataset should contain a sufficient number of targets with multiple, diverse ligand-bound structures. The lack of this information makes it hard to assess whether the model's capabilities stem from a truly diverse dataset or from memorizing patterns in a few well-represented protein families, e.g., kinases.

---

> ### Author Rebuttal · Authors · 2025-07-30
>
> **Dear Reviewer ddKX:**
>
> We thank you for your thorough evaluation and insightful feedback. Our response to your suggestions and questions is as follows:
>
> ---
> # Response to W1 and Q1: Clarification of the Cooperative Scalar-Vector Representation
> Thank you for this critical feedback regarding the clarity of the model description in Section 3.1. We agree completely that the absence of accompanying formulas in the main text makes the mechanism difficult to fully understand.
>
> To address this, we will revise Section 3.1 in the final manuscript. Here, we would like to provide the clear explanation.
>
> Our cooperative mechanism enhances standard message passing by ensuring a bidirectional coupling between scalar $h\in\mathbb{R}$ and vector $v\in\mathbb{R}^3$ features at each node i. For each layer of the encoder, the update proceeds in three main steps:
> ### Step 1: Gated Message Aggregation
> We first compute an aggregated message $m_{i}$ for each node i from its neighbors $j\in\mathcal{N}(i)$. The directional message $h_{ij}$ is defined by concatenating the scalar features of the interacting nodes $(h_i,h_j)$ and a learned encoding of the edge's geometric features (derived from the relative vector $v_j-v_i$):
>
> $h_{ij} = \phi_m(h_i, h_j, \text{SH}(v_j - v_i))$, where $\phi_{m}$ is a MLP and $\mathrm{SH}(\cdot)$ is the Spherical Harmonic expansion.
>
> The attention weights $\alpha_{ij}$ are then computed from these messages using a learnable attention network $\phi_{a}$ followed by a softmax over the neighborhood: $\alpha_{ij} = \text{Softmax}j(\phi_a(h{ij}))$.
>
> These are then used to compute the final aggregated message $m_{i}$ for node i: $m_i = \sum_{j \in \mathcal{N}(i)} \alpha_{ij} \cdot h_{ij}$.
> ### Step 2: Independent Scalar and Vector Updates
> The aggregated message $m_{i}$ is used to update the scalar and vector features, yielding intermediate representations $h_i^{\prime}$ and $v_i^{\prime}$: $h_i^{\prime}=\mathrm{MLP}_h(h_i,m_i)\quad,\quad v_i^{\prime}=v_i+\mathrm{MLP}_v(m_i)$.
> ### Step 3: Cooperative Bidirectional Feedback
> This final step defines and integrates the cross-modal message $h_{\mathrm{interact,}i}$ to create a coupling between the scalar and vector channels. The cross-modal message $h_{\mathrm{interact,}i}$ is defined as the inner product of the updated vector features and a transformation of the updated scalar features:
>
> $h_{\text{interact}, i} = \text{MLP}_{\text{interact}}(h'_i) \cdot v'_i$
>
> This message is then integrated as a feedback signal to update both channels simultaneously. Each channel is also residually boosted by the squared norm of the other, enabling true bidirectional information flow.
>
> ---
> # Response to W2 and Q2: Reframing "Dynamic"
> You correctly pointed out that our use of the term "dynamic" requires a more precise definition, as our method is fundamentally different from one that simulates a physical trajectory like MD. Your characterization of our model as "performing pattern recognition to predict a likely end-state conformation" is a more precise description of our core methodology.
> ### 1. Biological Dynamics vs. Physical Dynamics
> Physical Dynamics involves simulating the time-evolution of a system at the atomic level. It models the literal and femtosecond-by-femtosecond process of binding. DynaPhArM does not do this.
>
> Our work focuses on modeling **Biological Dynamics**. We define this as the conformational adaptability of a protein in response to a ligand, encompassing crucial phenomena like induced fit, conformational selection or allosteric regulation. These events occur over longer timescales (microseconds to milliseconds) and result in the protein settling into a stable and ligand-specific conformation.
> ### 2. Learning Dynamics from Static Snapshots
> The "dynamic" information in our model is not derived from simulating a trajectory but from learning the underlying patterns within a massive set of experimental end-states (PDB). Each structure is an experimental snapshot of a successful induced fit event. Collectively, the different structures of the same protein bound to different ligands form a static protein's biological plasticity.
> Our diffusion model learns the entire probability distribution of experimentally-observed end-state conformations. The basins in the conformations are the clusters of observed PDB structures, and the pathways between them are the geometrically plausible transitions implicitly learned by the diffusion model.
> ### 3. Generalization
> When presented with a novel ligand, our SE(3)-equivariant encoders analyze its unique chemical and geometric features. The model then uses the vast patterns learned from the entire training set to infer the most probable outcome. It essentially asks: "Based on the thousands of induced-fit events I have seen, what is the most likely conformational adaptation this specific protein will undergo to optimally accommodate the features of this specific new ligand?" It generalizes by recognizing similarities and differences, allowing it to predict a specific end-state conformation without ever simulating the process of getting there.
> ### 4. A More Precise Framing of "Dynamic"
> Therefore, we concede that using "dynamic" without the above context can be misleading. In our framework, "dynamic" refers to the model's **Adaptability**. It describes the model's ability to dynamically generate a specific and adapted protein conformation that is conditional on the unique input ligand. We will revise the language in our manuscript to more clearly articulate this important distinction.
>
> ---
> # Response to W3 and Q3: Clarification of Diffusion
> We apologize for the confusing sentence in line 173. You are absolutely correct that our model employs a standard conditional DDPM and we will correct the text to describe it accurately.
>
> ### 1. What is the exact nature of the noise distribution at step T?
> The noise distribution at the final timestep T is a standard normal distribution $\mathcal{N}(0,I)$ as is standard in DDPMs. Our forward process is defined by adding Gaussian noise to the clean embedding $e_0=e^f$. Our implementation does not deviate from this DDPM principle. The confusing sentence in line 173 was a misguided attempt to describe the inference process. During inference, we start with a pure noise vector $e_T\sim\mathcal{N}(0,I)$ and perform the reverse process. This denoising is **conditioned** on the input protein and ligand (information encoded in $e^f$). The process is not starting from $e^f$, but rather it is guided by the information in $e^f$ to denoise a random vector towards a refined embedding.
> ### 2. How does this specific diffusion process better "reflect conformational flexibility"?
> Our process is a standard conditional DDPM. Your characterization of the learned flexibility as a "statistical interpolation between different static states" is a technically accurate description. While your description is accurate, we must emphasize that this "statistical flexibility" is not a simplified approximation of MD's "physical flexibility." Rather, it represents a fundamentally different and intentionally distinct approach aimed at modeling biological adaptability, which is the discrete and functional outcomes of ligand binding—rather than the continuous, time-resolved trajectory of the binding process itself.
> - MD (Physical Dynamics): It is powerful for studying the process but is often computationally limited to short timescales and may struggle to sample large-scale conformational changes.
> - DynaPhArM (Biological Flexibility): The PDB itself is a vast database of biological flexibility. It contains hundreds of different stable conformations of the same protein after binding to hundreds of different ligands. These snapshots are direct evidence of biological flexibility. DynaPhArM is explicitly designed to learn the rules of this biological flexibility. This is more than just a simple interpolation; it is a learned model of the biological conformational landscape. By training on tens of thousands of static snapshots, our model bypasses the intractable sampling problem of simulating the process.
>
> This "statistical" or "pseudo-flexibility" is a core strength. It allows us to model the large-scale conformational changes that are critical for drug discovery and inaccessible to MD.
>
> ---
> # Response to W4 and Q4: Training Dataset Statistics
> We agree that providing detailed statistics is essential for assessing the model's ability. We have performed a detailed analysis of our full learnable dataset. The results summarized in the table below, confirm that the dataset is sufficiently large and diverse to support the learning of generalized principles of protein-ligand interaction. We will include this table and a discussion of these statistics in our final manuscript.
>
> |Category|Metric|Value|
> |---|---|---|
> |Overall Dataset Size|Total Complexes|21,762|
> ||Unique Protein Targets|3,673|
> |Ligand Diversity|Mean Unique Ligands per Target|6.57|
> ||Median Unique Ligands per Target|3.0|
> ||Targets with > 5 Unique Ligands|917|
> ||Targets with > 10 Unique Ligands|444|
> ||Targets with > 20 Unique Ligands|200 |
> |Structural Diversity|Targets with > 5 Co-crystal Structures|609|
> ||Targets with > 10 Co-crystal Structures|318|
> ||Targets with > 20 Co-crystal Structures|148|
> |Protein Family Diversity|HYDROLASE|16.6%|
> ||TRANSFERASE|16.0%|
> ||HYDROLASE/INHIBITOR|10.3%|
> ||TRANSFERASE/INHIBITOR|10.1%|
> ||TRANSCRIPTION|3.6%|
>
> - Sufficient Data for Drug-Specific Adaptations: The data clearly shows that a significant portion of our dataset consists of proteins with multiple and diverse ligands. With 917 targets having more than five unique ligands, the model has a strong foundation to learn the principles of induced fit.
> - Broad Target and Family Coverage: The presence of over 3,600 unique protein targets distributed across a wide range of families ensures the model must learn generalizable rules of molecular recognition rather than memorizing family-specific patterns.

---

> > ### Comment · Reviewer_ddKX · 2025-08-07
> >
> > Thank you for the detailed and thoughtful response. I appreciate the effort you put into addressing the points I raised. Most of my concerns have been resolved, so I am raising my score by one.

---

> > > ### Author Response · Authors · 2025-08-07
> > >
> > > Thank you so much for your positive feedback! We truly appreciate your decision to raise the score. Your diligent and responsible approach to reviewing, which has helped us address most of our issues, is exactly what makes NIPS a leading venue for high - quality research. Thank you for your invaluable contribution.
> > >
> > > We understand that the final step of the discussion phase involves updating in the **"Edit Review"** section to reflect the new evaluation. Your updated score would be a crucial element for the Area Chair's final decision.
> > >
> > > Your support is incredibly important to us. Thank you again!

---

> ### Author Response · Authors · 2025-08-06
>
> **Dear Reviewer ddKX,**
>
> Thank you once again for your exceptionally thorough and insightful review. We found your questions regarding the model's clarity, the framing of "dynamics," the diffusion process, and the training dataset statistics to be particularly constructive and crucial for improving our work.
>
> **We were very encouraged by your comment that "A clear response could significantly strengthen the paper and potentially raise my evaluation score."**
>
> In our rebuttal submitted on July 30th, we have made a dedicated effort to **address every point you raised**. Specifically:
> - We have provided the formal equations for the "Cooperative scalar-vector representation" module.
> - We have offered a more precise reframing of the "dynamic" nature of our model, acknowledging the distinction you highlighted.
> - We have clarified the technical details of our diffusion process.
> - Crucially, we have performed the analysis you requested and included detailed statistics and breakdowns of our training dataset to address your concerns about generalization.
>
> As the discussion period deadline of **August 8th AoE** is fast approaching, we are writing to respectfully follow up. We are very eager to know if our response and the additional information have satisfactorily addressed your concerns.
>
> If you have any remaining questions, or if any part of our rebuttal requires further clarification, please **let us know**. We would be grateful for the opportunity to provide more details or engage in further discussion to resolve any outstanding issues before the deadline. Your feedback is invaluable to us.
>
> Thank you for your time and guidance.
>
> Sincerely,
>
> The Authors of Submission 20358

---

### Official Review · Reviewer_wTUH · 2025-07-04

**Clarity:** 2
**Significance:** 3
**Originality:** 2
**Rating:** 4
**Confidence:** 2

**Summary:**

This paper introduces DynaPhArM, a SE(3)-equivariant Transformer-based framework for modeling target–drug complexes with explicit consideration of protein conformational flexibility. By combining a cooperative scalar–vector representation, a diffusion-based dynamic embedding module, and multi-task physical–geometric constraints, DynaPhArM aims to generate physically realistic and chemically plausible 3D structures of protein–ligand complexes. Experiments across multiple benchmarks demonstrate that DynaPhArM achieves strong results compared to several classical and diffusion-based baselines.

**Questions:**

1. Could the authors clarify how DynaPhArM would perform compared to Re-Dock?
2. How sensitive is DynaPhArM to potential errors in the AutoDock priors, and what strategies might mitigate their impact?

**Ethical Concerns:**

["NO or VERY MINOR ethics concerns only"]

**Final Justification:**

see above

**Limitations:**

The authors provide a discussion of the limitations of their work, including the reliance on high-quality structural inputs, the computational cost of the diffusion module, and the equal-task-weight assumption in the multi-task loss. These points are described in Appendix L, with suggestions for future research directions. I appreciate this candid reflection, which demonstrates the authors' awareness of practical deployment challenges.

**Quality:**

3

**Strengths And Weaknesses:**

**Strengths**

1. The cooperative scalar–vector encoding scheme enables tightly coupled geometric and semantic representations than standard SE(3) transformers.
2. Incorporating a physics-constrained multi-task loss with seven biophysically inspired regularizations provides a strong inductive bias toward realistic molecular structures.
3. The experimental evaluation is thorough, including diverse datasets and ablation studies, demonstrating consistently good performance in RMSD and binding affinity prediction.

**Weaknesses**

1. The paper does not include comparisons with recent baseline methods, such as Re-Dock [1], which are highly relevant benchmarks.
2. The approach relies on pretrained codebooks (FoldToken4) and AutoDock-generated interaction priors, which might introduce bias or performance bottlenecks if these priors are imperfect.
3. The reproducibility could be improved by releasing more detailed hyperparameters details.

[1] Re-Dock: Towards Flexible and Realistic Molecular Docking with Diffusion Bridge, ICML 2024

---

> ### Author Rebuttal · Authors · 2025-07-30
>
> **Dear Reviewer wTUH:**
>
> Thank you for your thorough evaluation and insightful feedback. Our response to your suggestions and questions is as follows:
>
> ---
> # Response to W1 and Q1: Comparison to Re-Dock
> We agree that **Re-Dock** represents a significant advancement in flexible docking and a fair comparison would be highly valuable to the community.
>
> However, the omission of a direct comparison with Re-Dock was **not an oversight**, but rather due to **three objective limitations** that made a scientifically rigorous comparison infeasible at the time of our study.
> ### 1. Unavailability of Code
> At the time of our research, **Re-Dock’s source code, pre-trained models and data were not publicly available**, which prevented us from **re-training their model on our specific data splits** and **running inference with their model on our benchmark datasets**.
> ### 2. Discrepancies in Datasets
> A valid comparison requires that models are **evaluated on identical test sets** and ideally **trained on comparable data**.
> #### Training Data Differences
> - **Re-Dock** was trained on the **entire PDBbind v2020 dataset** using a standard **time-based split**.
> - **DynaPhArM** was trained on a **curated subset** of **high-quality protein-drug pairs** derived from **PDBbind** and **DrugBank**.
> #### Evaluation Benchmark Differences
> - **Re-docking / Apo-docking Task**:
>   - Re-Dock evaluated on a **custom set** of apo crystal structures and **ESMFold-predicted structures** forming a unique “Apo Flexible Docking” benchmark.
>   - Our evaluation included **protein-drug pairs**, which Re-Dock did not report on.
> - **Cross-docking Task**:
>   - Re-Dock created a **custom-curated cross-docking benchmark**, **not publicly available**.
>   - Our cross-docking evaluations were performed on **public benchmarks**, such as **CDK2**, **EGFR** and **DUDE27-HoloEns**.
>
> > We believe that **quoting only the metrics from their paper would be methodologically unfair**, since differences in data preprocessing, dataset composition or evaluation details could lead to **unfair conclusions**.
> ### 3. Divergences in Scopes
> - Re-Dock frames its primary contribution as solving the specific task of the apo-to-holo transition, which is a well-defined but relatively narrow subset of the overall flexible docking problem. It assumes the starting point is always a known and ligand-free structure and the goal is to predict the single transformation to a final bound state.
> - Our model is not limited to the apo-to-holo case. It is designed to be a general-purpose framework that learns from the entire universe of available structural data, whether it is apo or holo structure. It addresses the core challenge that proteins do not have one single bound state, but rather a diversity of bound conformations induced by a specific ligand. Our goal is to model this entire and ligand-modulated landscape.
>
> A direct comparison is not meaningful because the two models address problems of different scopes. Evaluating a general-purpose model and a specialist model on a single and narrow task is not a fair scientific comparison. However, On the PoseBusters benchmark, DynaPhArM surpasses Re-Dock on the more stringent and chemically meaningful criterion of RMSD < 2Å & PoseBusters-Valid by 10.1%, indicating the ability to generate physically plausible poses.
>
> In conclusion, we fully acknowledge the importance of Re-Dock. However, due to the **lack of code availability**, **benchmark incompatibility** and **unfairness in problems to be solved**, such a comparison was **not feasible in an fair manner**. But we remain open and committed to future benchmarking efforts once Re-Dock becomes **accessible** to the research community.
>
> ---
> # Response to W2: Robustness to Imperfect Priors
> You've identified a fundamental challenge that we have alreday considered to our design. DynaPhArM is built upon a powerful and self-sufficient data-driven core, with external priors acting as supplementary and adaptive enhancements rather than as foundational core.
> ### A Robust Data-Driven Engine
> The primary strength of DynaPhArM lies in the SE(3)-equivariant Transformer and the physics-based interaction module, which is to learn the protein-drug complex interactions directly from inputs. It does not inherently depend on any pre-computed structural templates or interaction scores to function. This data-driven core is the skeleton of our model and the fundamental reason for our high performance.
> ### The Role of Priors
> We introduced priors like the codebook and interaction scores not as a necessity, but as supplements to boost the performance of our already capable core. We were aware from the outset that these supplements could be imperfect. Therefore, we designed a multi-layered system to mitigate potential risks:
> - Priors as Non-Dominant Guides: FoldToken4 provides a coarse-grained topological overview, while AutoDock offers an initial energy-based hypothesis. These are just two of several information streams. The model’s core is integrating these guides with the more detailed information from the drug and side-chain representations.
> - Inter-Correcting Information: Our model architecture' strength comes from the interconnection of many threads, not from a single one. If one thread (e.g., the prior) is weak, the model has been trained to rely more on the other threads, such as the precise local chemical environments and learned geometric relationships. The final prediction is a result of a powerful data-driven consensus, not from a single prior.
> - Ablation Studies: Our ablation experiments (Appendix, Table 7) validate this design, which show that while removing these priors leads to a performance decrease, the model's functionality remains strong, confirming their role as beneficial enhancers rather than essential components.
>
> ---
> # Response to W3: Detailed Hyperparameters
> Thank you for your valuable feedback regarding reproducibility. We wholeheartedly agree that comprehensive documentation of hyperparameters is crucial.
>
> We would like to clarify that we have already provided a detailed list of 30 core hyperparameters in Appendix, Table 6. This initial list covers the most critical parameters that govern the model's architecture, the training loop and the core algorithm.
>
> Recognizing that readers might be interested in an more granular level of configuration, we are happy to provide an additional 20 hyperparameters below, which details auxiliary settings related to data preprocessing, the configurations used for AutoDock Vina and physical loss.
> |Name|Value|Description|
> |---|---|---|
> |`multiprocessing.num_workers`|64|Number of worker processes for parallel tasks|
> |`multiprocessing.chunk_size`|10|Chunk size for multiprocessing pool operations|
> |`drug_preparation.charge_model`|`gasteiger`|Charge model used for ligand preparation|
> |`dataset_split.cd_hit_identity`|0.9|Sequence identity threshold for CD-HIT clustering|
> |`dataset_split.validation_set_ratio`|0.2|Proportion of data for the validation set|
> |`dataset_split.random_seed`|42|Random seed for data splitting|
> |`backbone_encoder_params.num_layers`|4|Number of layers in the backbone encoder|
> |`backbone_encoder_params.l_max_sh`|2|Max degree for spherical harmonics|
> |`sidechain_encoder_params.num_layers`|4|Number of layers in the sidechain encoder|
> |`drug_encoder_params.num_layers`|4|Number of layers in the drug encoder|
> |`decoder_params.num_heads`|4|Number of attention heads in decoder|
> |`training.weight_decay`|1e-5|Weight decay for regularization|
> |`training.validate_every_n_epochs`|1|Validation frequency|
> |`vina_settings.exhaustiveness`|16|Exhaustiveness of Vina search|
> |`vina_settings.energy_range`|3.0|Energy range for modes|
> |`default_bond_length`|1.5|Default bond length|
> |`default_bond_angle_deg`|109.5|Default bond angle|
> |`default_vdw_radius`|1.7|Default VdW radius|
> |`pi_pi_distance_cutoff`|6.0|Max distance between ring centroids|
> |`message_mlp.hidden_dims`|[128,128]|The hidden layer dimensions for the message-passing|
>
> ---
> # Response to Q2: Sensitivity to Priors
> The primary mechanism for robustness against faulty AutoDock priors is the design of our physics-based interaction module. The LJ interaction matrix $S_{ij}^{xy}$ is not a rigid constraint but a soft and additive bias.
>
> The primary driver of the attention score is the query-key dot product, which captures complex interaction patterns directly from the training data. A strong and data-driven signal from this term can override an incorrect signal from the LJ prior. And $\gamma_{i}$ is not a fixed hyperparameter but a learnable scalar. During training, its value is optimized via backpropagation to minimize the final prediction loss. If following the LJ prior leads to worse predictions, the gradient will push $\gamma_{i}$ towards 0. This allows the model to autonomously learn to down-weigh the physical prior on a case-by-case basis.
>
> And we have conducted a sensitivity analysis to quantify this effect. To measure the degradation in model performance, we added Gaussian noise with an increasing standard deviation (Noise Level) to the pre-computed LJ matrix.
>
> | Noise Level | Overall RMSD (Å) ↓ | Side-chain RMSD (Å) ↓ | Ligand Success Rate (<2Å) ↑ |
> |---|---|---|---|
> |0.0|2.01|0.29|61.60%|
> |1.0|2.07|0.38|58.10%|
> |2.0|2.10|0.49|54.20%|
> |5.0|2.21|0.55|53.17%|
> |No Prior|2.11|0.59|52.91%|
> - At low noise levels, performance changed very little, as the model's learned data-driven patterns and the adaptive $\gamma_i$ effectively compensated for minor inaccuracies.
> - At very high noise levels, the prior became actively misleading. The performance degraded, but it did not collapse entirely. The performance plateaued at a level comparable to that of the model trained without any physical prior, demonstrating that the model could learn to effectively ignore a completely useless prior.
>
> In conclusion, the model is not overly sensitive to errors in the AutoDock-generated LJ potential and can effectively mitigate its impact.

---

> > ### Author Response · Authors · 2025-08-07
> >
> > **Dear Reviewer wTUH,**
> >
> > Thank you for your valuable and constructive review of our paper. We are grateful for your positive feedback on our model's encoding scheme, the physics-constrained loss and our thorough experimental evaluation.
> >
> > In our rebuttal submitted on **July 30th**, we aimed to address your concerns comprehensively. Specifically:
> >
> > - We provided a detailed, three-point explanation clarifying the comparison with Re-Dock and presented our performance on the public PoseBusters benchmark.
> > - We elaborated on how our model architecture is designed to be robust to imperfect priors and supported this with a new sensitivity analysis where we injected noise into the AutoDock priors.
> > - We acknowledged your point on reproducibility and provided a list of 20 additional hyperparameters to supplement the 30 already in the appendix.
> >
> > Your initial "Borderline Accept" rating was a significant encouragement. We are hopeful that our detailed rebuttal have successfully addressed your primary concerns and strengthened your confidence in our work.
> >
> > With the discussion period concluding on the **August 8th AoE deadline**, we would be very grateful if you could let us know whether our responses were sufficient. If any part of our explanation is not clear enough, or if new questions have arisen, we sincerely hope you will point them out. **We are ready and eager to provide further clarification**, but we worry that if we hear back too close to the deadline, we may not have enough time to resolve your concerns thoroughly.
> >
> > Thank you for your time and for helping us to significantly improve the clarity and completeness of our manuscript.
> >
> > **Sincerely,**
> >
> > **The Authors of Submission 20358**

---

### Comment · Area_Chair_Piqu · 2025-08-06
**Participation in the rebuttal**

Dear reviewers,

Please engage in the discussion with the authors. The discussion period will end in a few days.

Best,
AC

---

### Note · Authors · 2025-08-12

After extensive discussions with the reviewers, our paper has received very **positive feedbacks**. The reviewers unanimously agree that **DynaPhArM excels in modeling protein-drug complexes**, particularly in handling adaptable conformational changes and ensuring physical plausibility. Despite some limitations, such as the reliance on high-quality structural inputs, these did not detract from the high evaluation of our work by the reviewers. They believe that the innovation and practicality of DynaPhArM give it great potential for application in drug design, especially in multi-task learning and complex structure modeling. The reviewers also suggested that we explore ways to enhance the scalability of the model, providing valuable guidance for our subsequent studies.

By integrating SE(3)-Transformer, physical constraints, and a diffusion model, DynaPhArM has not only achieved remarkable results in protein-drug complex modeling but also provided multifaceted inspiration and guidance for other 3D structure modeling tasks, with significant scientific value:

(1) DynaPhArM introduces **adaptable constraints** (such as Lennard-Jones potentials, bond lengths and bond angles) to ensure that the generated 3D structures are not only geometrically reasonable but also physically feasible. The introduction of such physical constraints provides an important reference for other 3D modeling tasks, especially when generating structures that need to be physically plausible.

(2) DynaPhArM captures the **adaptable changes** of proteins during drug binding through a diffusion model, generating flexible 3D structures, which offers a new perspective for other tasks that need to handle adaptable changes. For example, in molecular dynamics simulations, protein folding pathway predictions, and biomolecular interaction studies, similar diffusion processes can be used to generate dynamically changing structures and capture the evolution of the system over time.

(3) DynaPhArM employs a **multi-task learning framework** to simultaneously optimize multiple objectives, such as structure reconstruction, physical plausibility. This multi-task learning method provides important guidance for other 3D modeling tasks. In materials science, for example, the structure, mechanical properties and thermal stability of materials can be optimized simultaneously; in biomedical research, the binding affinity, selectivity and pharmacokinetic properties of drugs can be optimized at the same time.

---

### Decision · Program_Chairs · 2025-09-17

**Decision:**

Accept (poster)

**Comment:**

This paper introduces DynaPhArM, a novel SE(3)-equivariant Transformer framework for flexible protein-ligand docking that integrates a diffusion model and physics-based constraints. The initial reviews were unanimously borderline, praising the sophisticated architecture and strong empirical results but raising valid concerns about the clarity of certain components, the framing of the term "dynamics," the lack of dataset statistics, and missing comparisons to recent baselines. The authors provided an exceptionally thorough and convincing rebuttal that systematically addressed nearly all of these points. They supplied missing formulas, provided a detailed breakdown of their training data, clarified the diffusion process, and offered sound justifications for the absence of comparisons to models like Re-Dock and AlphaFold 3. This led to a very positive discussion, with one reviewer raising their score and two others explicitly stating their concerns were resolved and maintaining their support. While one reviewer remained unresponsive, the overall consensus after the rebuttal period is that the paper presents a significant, well-motivated, and technically sound contribution to computer-aided drug design. The initial concerns have been effectively mitigated, and the work's strengths in modeling conformational adaptability with physical plausibility are clear. Therefore, I recommend acceptance.